# A tension-adhesion feedback loop in plant epidermis

**Stéphane Verger\*, Yuchen Long, Arezki Boudaoud, Olivier Hamant\***

Laboratoire de Reproduction et Développement des Plantes, Université de Lyon, ENS de Lyon, UCB Lyon 1, CNRS, INRA, Lyon, France

**Abstract** Mechanical forces have emerged as coordinating signals for most cell functions. Yet, because forces are invisible, mapping tensile stress patterns in tissues remains a major challenge in all kingdoms. Here we take advantage of the adhesion defects in the *Arabidopsis* mutant *quasimodo1 (qua1)* to deduce stress patterns in tissues. By reducing the water potential and epidermal tension *in planta*, we rescued the adhesion defects in *qua1*, formally associating gaping and tensile stress patterns in the mutant. Using suboptimal water potential conditions, we revealed the relative contributions of shape- and growth-derived stress in prescribing maximal tension directions in aerial tissues. Consistently, the tension patterns deduced from the gaping patterns in *qua1* matched the pattern of cortical microtubules, which are thought to align with maximal tension, in wild-type organs. Conversely, loss of epidermis continuity in the *qua1* mutant hampered supracellular microtubule alignments, revealing that coordination through tensile stress requires cell-cell adhesion.

## Introduction

As our understanding of the role of forces in development deepens, assessing accurate stress patterns in tissues has become increasingly important (*Roca-Cusachs et al., 2017*). Stress patterns can be revealed through three approaches: 1- Computational models, for example with spring networks or finite elements, with relevant assumptions on tissue mechanics for animal (e.g. *Sherrard et al., 2010*) and plant (e.g. *Bozorg et al., 2014*) systems, 2- Strain measurements following local cuts at the subcellular (e.g. *Landsberg et al., 2009*) or organ (e.g. *Dumais and Steele, 2000*) scale, 3- Strain measurement of deformable objects (e.g. FRET-based molecular strain sensors [*Freikamp et al., 2017*], oil microdroplets [*Campàs et al., 2014*], elastomeric force sensors [*Wolfenson et al., 2016*]). Previous work on animal single cells showed that hyperosmotic media can affect membrane tension and thus the molecular effectors of cell migration, like actin filaments, RAC activity or WAVE complex, suggesting that the corresponding mutants could be rescued by a modification of the osmotic conditions of the medium (*Batchelder et al., 2011*; *Houk et al., 2012*; *Asnacios and Hamant, 2012*). Consistently, adding sorbitol in growth media is sufficient to rescue defects in yeast endocytic mutants (*Basu et al., 2014*). Here we take inspiration from these single cell studies and apply the same logic at the multicellular scale. Using an *Arabidopsis* mutant with severe cell adhesion defects, we partially rescue these defects by modifying the water potential of the growth medium and we deduce the maximal direction of tension in tissues from the gaping pattern following growth, without any external intervention.

In plants, cell adhesion is achieved through the deposition of a pectin-rich middle lamella between contiguous cell walls (*Orfila et al., 2001*; *Daher and Braybrook, 2015*; *Willats et al., 2001*; *Chebli and Geitmann, 2017*; *Jarvis et al., 2003*; *Knox, 1992*). QUASIMODO1 (QUA1) encodes a glycosyltransferase that is required for pectin synthesis and cell adhesion (*Bouton et al., 2002*; *Mouille et al., 2007*). Here we reasoned that the resulting cell-cell gaps may in principle reveal the stress pattern in tissues. Yet, it has not been formally demonstrated that gap opening could be

**\*For correspondence:**
stephane.verger@ens-lyon.fr (SV);
Olivier.Hamant@ens-lyon.fr (OH)

**Competing interests:** The authors declare that no competing interests exist.

**eLife digest** The parts of a plant that protrude from the ground are constantly shaken by the wind, applying forces to the plant that it must be able to resist. Indeed, mechanical forces are crucial for the development, growth and life of all organisms and can trigger certain behaviours or the production of particular molecules: for example, forces that bend a plant trigger gene activity that ultimately makes the stem more rigid.

Mechanical forces can also originate from inside the organism. For example, the epidermal cells that cover the surface of a plant are placed under tension by the cells in the underlying layers of the plant as they grow and expand. The exact pattern of forces in the plant epidermis was not known because they cannot be directly seen, although scientists have tried to map them using theoretical and computational modeling.

A mutant form of the *Arabidopsis* plant is unable to produce some of the molecules that allow epidermal cells to adhere to each other. Verger et al. placed the mutants in different growth conditions that lowered the pressure inside the plant, and consequently reduced the tension on the epidermal cells. This partly restored the ability of epidermal cells to adhere to each other, although gaps remained between cells in regions of the plant that have been predicted to be under high levels of tension. Verger et al. could therefore use the patterns of the gaps to map the forces across the epidermis, opening the path for the study of the role of these forces in plant development.

Further experiments showed that cell adhesion defects prevent the epidermal cells from coordinating how they respond to mechanical forces. There is therefore a feedback loop in the plant epidermis: cell-cell connections transmit tension across the epidermis, and, in turn, tension is perceived by the cells to alter the strength of those connections.

The results presented by Verger et al. suggest that plants use tension to monitor the adhesion in the cell layer that forms an interface with the environment. Other organisms may use similar processes; this theory is supported by the fact that sheets of animal cells use proteins that are involved in both cell-cell adhesion and the detection of tension. The next challenge is to analyse how tension in the epidermis affects developmental processes and how a plant responds to its environment.

related to tissue tension. Furthermore, the severe defects in the mutant make it hard to deduce a stress pattern in such distorted tissues. We thus developed a protocol amenable to partially rescue the adhesion defect through water potential modulation, allowing us to relate adhesion to tissue tension on the one hand, and deduce a pattern of stress in various plant tissues on the other hand. This mutant also allowed us to investigate how the loss of adhesion affects the propagation of mechanical stress and thus tension-dependent cell-cell coordination.

## Results

### Adhesion defects in *qua1* mutants depend on the water potential of the growth medium

The *quasimodo1 (qua1)* and *qua2* mutants, respectively mutated in a galacturonosyltransferase and a pectin methyltransferase, are both required for the synthesis of a fraction of the cell wall pectins. They also display a comparable cell adhesion defect phenotype (*Bouton et al., 2002*; *Mouille et al., 2007*). For practical reasons, all the work reported in this study was performed with *qua1-1* (WS-4 background), although we observed similar phenotypes in the *qua2-1* mutant (Col-0 background). Because the *qua1* mutant is very sensitive to sucrose in the medium, which leads to metabolic stress and growth arrest of the seedling (*Gao et al., 2008*), we grew the seedlings on a medium containing no sucrose to focus on the cell adhesion phenotype. In these conditions, we could observe cell separation in the epidermis of hypocotyls, stems, cotyledons, and leaves (*Figure 1E*; *Figure 1—figure supplement 1*), consistent with the epidermal theory of growth where the epidermis is put under tension through the pressure exerted by inner tissues, and thus is load-bearing for aerial organs (*Kutschera and Niklas, 2007*; *Savaldi-Goldstein et al., 2007*; *Maeda et al., 2014*).

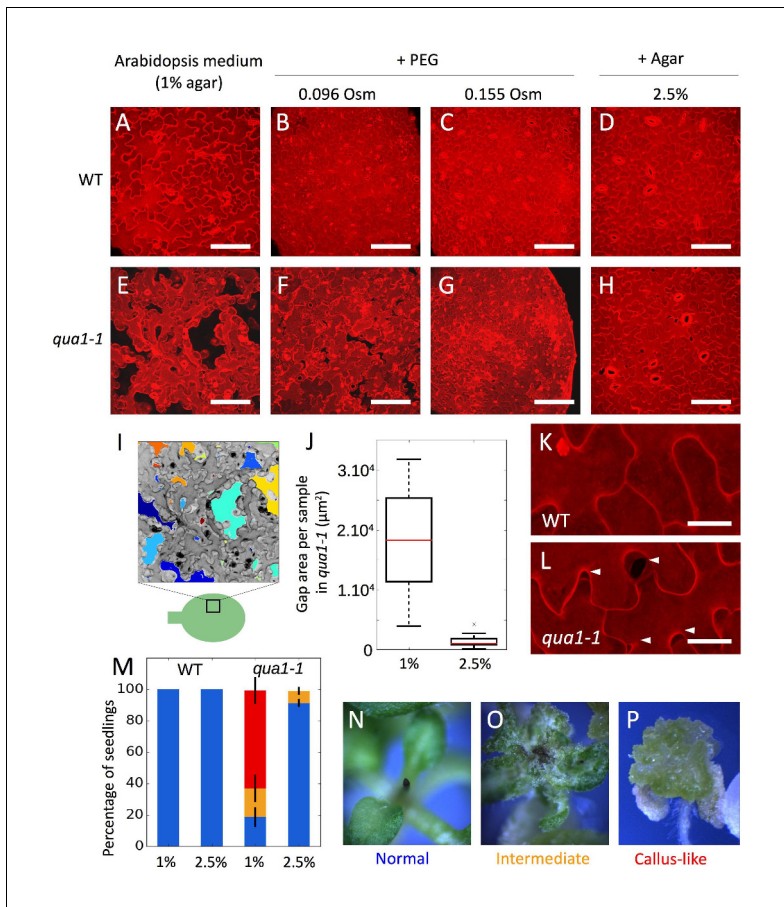

**Figure 1.** Adhesion defects in *qua1* mutant scale to the water potential of the growth medium. (**A–H**) Z-projections (maximal intensity) of confocal stacks from representative (12 samples observed in three biological replicates for each condition), propidium iodide stained, five days old cotyledons, showing the effect of the decreased medium water potential in the wild type (**A–D**) and the *qua1-1* mutant (**E–H**). A and E are standard growth condition. B-D and F-H are growth conditions with decreased water potential. B, F and C, G water potential is deceased with PEG, which increases the osmolarity of the medium of 96 mOsm (**B, F**) and 155 mOsm (**C, G**). In D and H water potential is decreased by increasing the agar concentration in the medium to 2.5%. (**I**) Representation of the semi automated image analysis process used to detect and quantify the area of gaps per sample shown in I (method further described in the material and method section and *Figure 1—figure supplement 4*). The upper part: Z-projections (maximal intensity) of a confocal stack in grayscale color and inverted pixel intensity corresponding to panel E. Cell separations are identified and labeled in multiple colors for visualization. The bottom part: schematic representation of a cotyledon and the relative position where Z-stacks were taken (square). (**J**) Box plot of the quantification of the total area of cell separation per image analyzed, in 1% and 2.5% agar growth conditions, and corresponding to the comparison of the panels E and H (12 samples quantified in three biological replicates for each condition, Welch's *t*-test p-value=0.0004). (**K–L**) Close-up from (**D**) and (**H**) respectively, showing the cell separations preferentially happening at the lobe-neck junction (Arrow heads), in the *qua1-1* mutant (**L**). (**M**) Phenotype of the shoot apex in the wild type and *qua1-1* mutant, grown on 1% and 2.5% agar medium (210 seedlings in three biological repetitions per condition, error bars represent the standard deviation over the three biological replicates). Color code is explained in panels N-P: normal stem and meristem (Blue, **N**), a callus-like apex (Red, **P**) or an intermediate phenotype (Orange, **O**). (**N–P**) Stereo microscope images of representative phenotypes as used for the quantifications shown in M. Scale bars (**A–H**), 100 µm. Scale bars (**K,L**), 20 µm.

The online version of this article includes the following figure supplement(s) for figure 1:

**Figure supplement 1.** Widespread cell adhesion defects in the *qua1-1* mutant.
**Figure supplement 2.** Seedling phenotypes in *qua1* mutant depend on the water potential of the growth medium.
**Figure supplement 3.** Dynamics of cell separation in *qua1-1* cotyledon epidermis.
**Figure supplement 4.** A semi automated cell separation image analysis pipeline.
*Figure 1 continued on next page*

*Figure 1 continued*

**Figure supplement 5.** Close-ups of cell-cell adhesion defects in *qua1-1*.

Because endogenous tensile stress in plant epidermis originates both directly and indirectly from turgor pressure, we next modified the water potential of the medium, reasoning that epidermal integrity should be restored in the *qua1* mutant if epidermal tension was decreased.

Increasing concentrations of Polyethylene Glycol (PEG) in the growth medium reduced the growth rate of wild-type and *qua1-1* seedlings, suggesting a reduction in turgor pressure *in planta* caused by a decrease in the water potential of the medium (*Figure 1—figure supplement 2A–C*). Strikingly, in the lowest water potential condition, the overall *qua1-1* phenotype was almost fully rescued (*Figure 1—figure supplement 2E–G*). Using propidium iodide staining and confocal imaging of the cotyledon pavement cells, we further confirmed that such osmotic conditions reduced cell separations in *qua1-1* (*Figure 1E–G*). Note that *qua1-1* pavement cells preferentially separated at the neck-lobe junction, consistent with previously calculated patterns of stress in this tissue (*Figure 1K and L*, [*Sampathkumar et al., 2014*], *Figure 1—figure supplement 3*, *Video 1* and *2*). This suggests that low tensile stress in the epidermis is sufficient to restore cell adhesion in the mutant.

Nonetheless, because PEG may have pleiotropic effects, we cannot exclude the possibility that the restoration of cell adhesion could be due to other factors. Therefore, we increased agar concentration in the medium, as an alternative way to affect water potential. Indeed, increasing the agar concentration reduces the water potential by decreasing its matrix potential (a component of the water potential, [*Owens and Wozniak, 1991*]) and in turn hinders the capacity of the plant to take up water, as does PEG-containing medium. As expected, we observed a similar restoration of the cell adhesion and seedling phenotype in *qua1-1*, to that observed on PEG-containing medium (*Figure 1H*, *Figure 1—figure supplement 2H*).

To go beyond these qualitative observations, we developed a semi-automated pipeline of image analysis amenable to identify individual gaps between cells, quantify their areas and their main orientations (see Material and methods, *Figure 1I*; *Figure 1—figure supplement 4*; *Verger and Cerutti, 2018*; copy archived at https://github.com/elifesciences-publications/

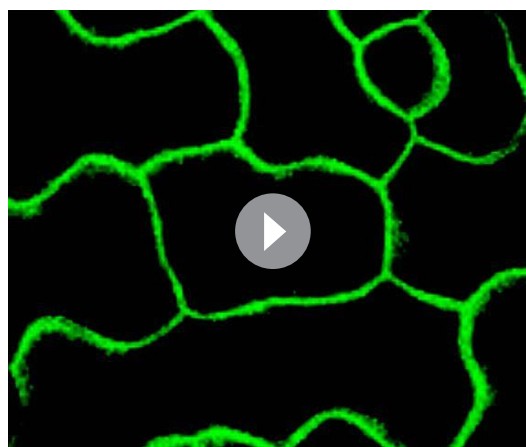

**Video 1.** Dynamics of cell separation in *qua1-1 pPDF1::mCit:KA1* cotyledon epidermis. Z-projections maximal intensity of confocal stacks taken for 72 hr, at 12 hr intervals.
https://elifesciences.org/articles/34460#video1

**Video 2.** Dynamics of cell separation in *qua1-1 pPDF1::mCit:KA1* cotyledon epidermis, high resolution close up of sample 2. Similar to *Figure 1—figure supplement 3A–D*. Z-projections max intensity of confocal stacks taken for 48 hr, at 12 hr intervals.
https://elifesciences.org/articles/34460#video2

Cell_separation_analysis). Based on images of five-day-old *qua1-1* cotyledons (*Figure 1I*), we found that, for a field of cells representing 138654 µm² per image, seedlings grown on 1% agar medium exhibited 17906 (±8955) µm² of cell separation per image (i.e. ca. 13% of the surface area, n = 12 samples), while seedlings grown on 2.5% agar medium displayed 1457 (±1140) µm² of cell separation per image (i.e. ca. 1% of the surface area, n = 12 samples; *Figure 1J*, Welch's *t*-test p-value=0.0004), confirming the rescuing effect of a low water potential on cell adhesion.

Last, to confirm that the propidium iodide staining truly reflected cell-cell adhesion defects, we analyzed the gaps in *qua1-1* at different stages with confocal microscopy and, at high resolution, with atomic force microscopy. Our images matched previously published SEM images of *qua1-1* mutants (*Bouton et al., 2002*), with stretched and detached outer walls at the cell-cell junction (*Figure 1—figure supplement 5*). Altogether, these results strongly suggest that adhesion defects in *qua1-1* indeed relate to the tensile status of the tissue.

## Adhesion defects in *qua1* mutants specifically relate to epidermal tension

At this stage, we find a correlation between the medium water potential and adhesion defects in *qua1-1*. To measure the impact on epidermal tension, we turned to atomic force microscopy (AFM) to obtain force-displacement curves on epidermal surfaces allowing us to extract a slope, which corresponds to the apparent stiffness of the material (*Figure 2*). We focused on cotyledons, as they are easier to manipulate under the AFM. To maintain tissue hydration, AFM live imaging was conducted in aqueous solutions: for 1% and 2.5% agar-grown cotyledons, cotyledons were submerged in water, while PEG-grown cotyledons were submerged in liquid Arabidopsis medium supplemented with mannitol to reach the same osmotic pressure as the PEG-infused medium (see Materials and methods). To measure cell-level mechanical properties over the epidermal surface, we performed indentations with an AFM probe much smaller than cell size (0.8 µm probe diameter compared to >10 µm cell width; *Figure 2A*). Approximately 10 ~ 15 µN indentations were performed to achieve 1 ~ 2 µm indentation depth (*Figure 2A*), deep enough to detect epidermal turgor pressure but relatively shallow compared to pavement cell thickness (typically 6 ~ 10 µm, [*Zhang et al., 2011*]). In these conditions, we are measuring the stiffness of single epidermal cells, and do not detect the stiffness of the rest of the tissue, notably the internal cell layers (*Beauzamy et al., 2015*).

As shown in *Figure 2*, results obtained from wild-type seedlings grown on medium supplemented with 155 mOsm PEG or 2.5% agar were similar. Similar trends were obtained in *qua1-1*, albeit with a globally reduced turgor pressure, apparent stiffness and cell wall tension (*Figure 2—figure supplement 1*). We also confirmed that the immersion medium had little impact on the measurement both in wild type and *qua1-1*, at least in the short term (*Figure 2—figure supplement 1*). First, we found that turgor pressure levels in the epidermis were not affected by a change in osmotic or matrix potential. This suggests that water potential primarily affect internal tissues and/or that the epidermis can osmoregulate efficiently, as already shown before (e.g. [*Shabala and Lew, 2002*]). Interestingly, when focusing on the outer wall, we found that apparent stiffness decreased by 15% in both PEG and high agar-grown seedlings, while cell wall tension in high agar and PEG-grown seedlings decreased by 18% and 16% respectively, thus demonstrating the impact of the corresponding treatments on epidermal tension. Although such differences in stiffness may appear small, comparable differences were obtained in other tissues, for example between central and peripheral zone of the shoot apical meristem, where differences in growth rates are of 200% to 300% (*Milani et al., 2014*). Furthermore, the concentrations of PEG and agar were chosen so as to maintain growth; in other words, differences could be stronger for higher concentrations, but these would not be relevant for our study. Altogether, these data formally relate the decreased water potential in the medium to a decrease in outer wall tension *in planta* and rescue of gaping patterns in *qua1-1*.

Because increasing agar concentration is not toxic to the cell, as shown by propidium iodide staining (see *Figure 1G and H*), we selected this protocol to alter water potential in the following experiments.

## Shape-derived tensile stress dominates at the stem apex

Both shape-derived stress and growth-derived stress contribute to the final pattern of stress in any given field of cells. Shape-derived stress, or pressure stress, is calculated based on the assumption

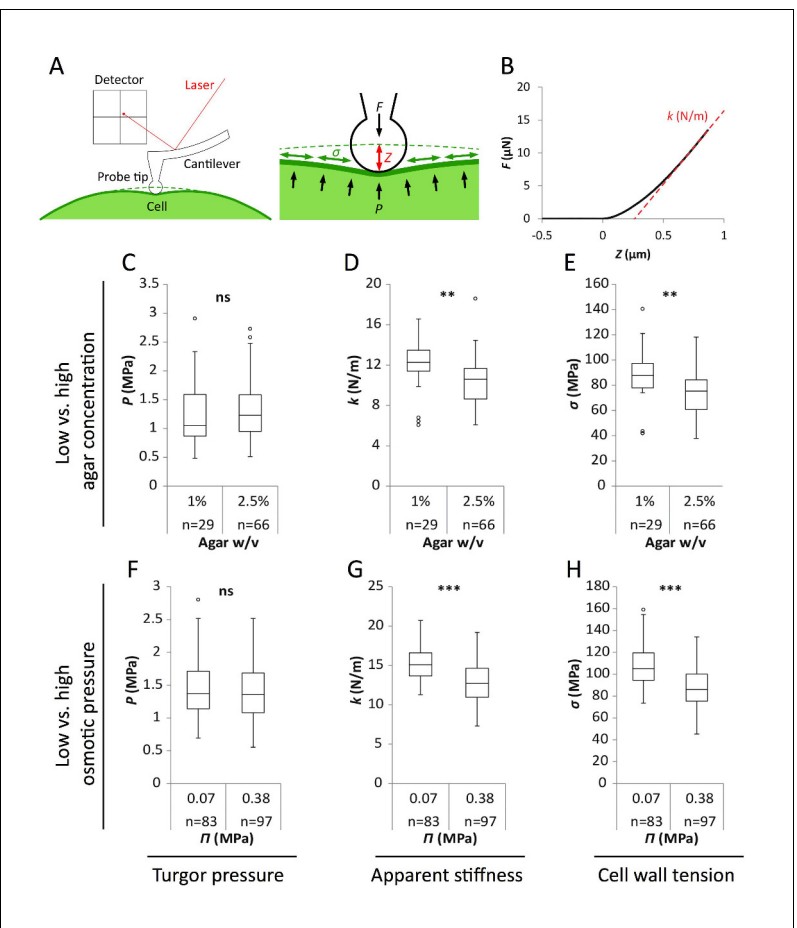

**Figure 2.** Reduced water potential in growth medium causes decrease in pavement cell stiffness and cell wall tension, not turgor pressure. (A) Schematic representation of AFM nano-indentation principle of measurements. $F$, indentation force; $Z$, indentation depth; $P$, turgor pressure; $\sigma$, cell wall tension. (B) Example of a typical AFM force curve (black line) and linear fit at deep indentation (red dotted line, 75 ~ 99% of maximum force) which depicts apparent stiffness $k$. (C–H) Box plots of the turgor pressure $P$ (C,F), apparent stiffness $k$ (D,G) and cell wall tension $\sigma$ (E,H) of cotyledon pavement cells grown on medium with differential agar concentration (1% and 2.5% w/v) (C–E) or osmotic pressure $\Pi$ (0.07 and 0.38 MPa) (F–H). Circles indicate Tukey's outliers. Student's $t$-test, ** indicates p-value<0.01; ***, p-value<0.001; ns, not significant; $n$, number of measured cells.

The online version of this article includes the following figure supplement(s) for figure 2:

**Figure supplement 1.** Controls for AFM measurements.

that an organ behaves like a pressure vessel, that is like a load-bearing envelope under tension. This is typically the case for individual plant cells (cell walls resist internal turgor pressure and thus are under tension, see for example *Sampathkumar et al., 2014*) for an example of stress prediction entirely based on cell shape) and aerial organs (in the epidermal theory of growth framework, (*Kutschera and Niklas, 2007*), see for example (*Hamant et al., 2008*) for an example of stress prediction entirely based on tissue shape) (*Figure 3N*). Growth-derived stress corresponds to mechanical conflicts arising from differential growth rates or directions ([*Rebocho et al., 2017*; *Thimann and Schneider, 1938*; *Kutschera, 1992*], *Figure 3N*). Using the *qua1-1* mutant, we dissected these two contributions to the global tensile stress patterns in various plant tissues.

First, we analyzed the tensile stress patterns in the inflorescence stem apex, that is where the cells are still actively dividing below the shoot apical meristem (*Figure 3A*). In normal in vitro growth condition, *qua1-1* seedlings often generate aberrant inflorescence meristems and stems, morphologically reminiscent of calli (see *Figure 1P*, and [*Krupková et al., 2007*]). To test whether this phenotype also correlates with the medium water potential, we used *in vitro* grown seedlings to

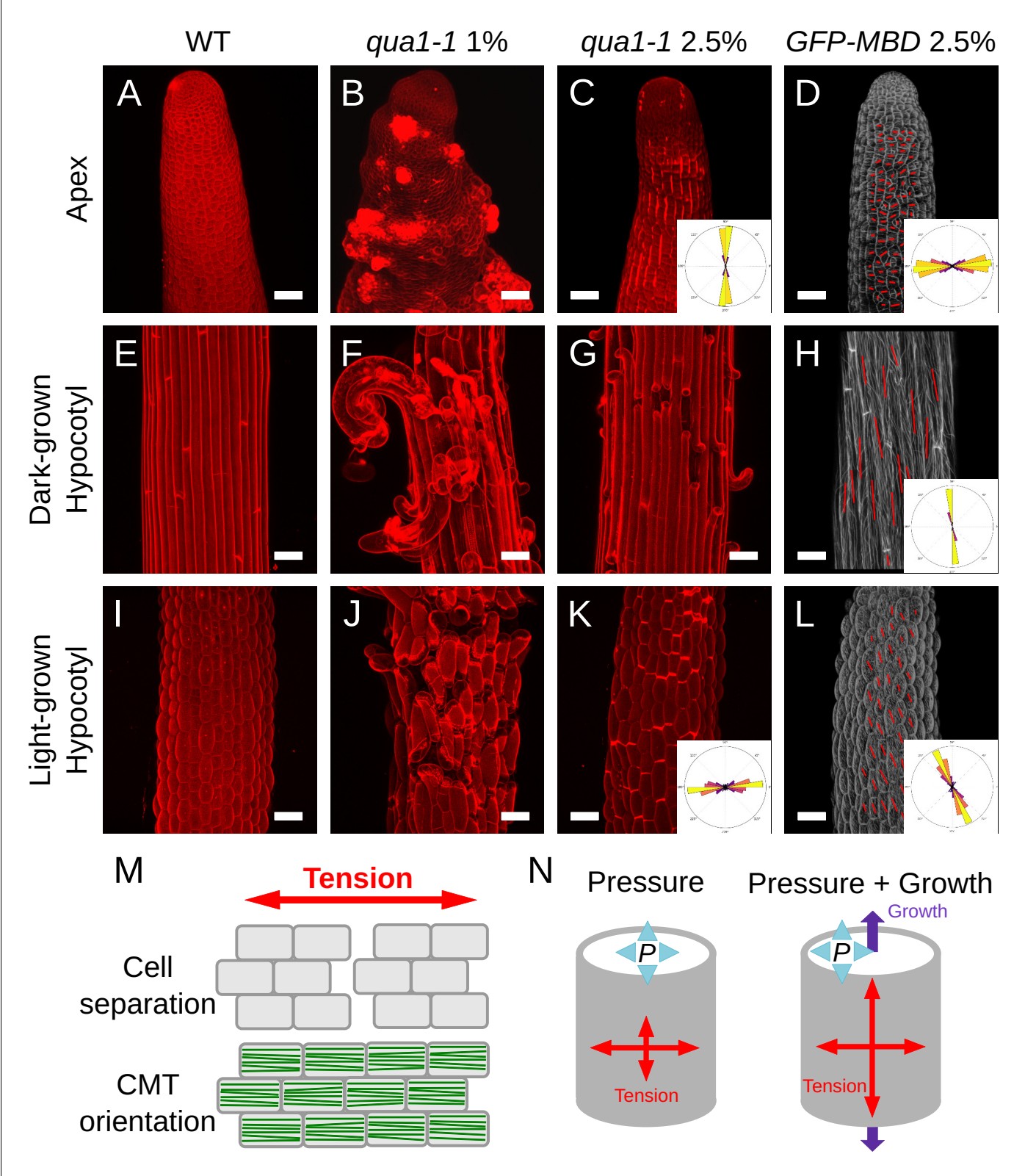

**Figure 3.** Both shape derived and growth derived tensile stress contribute to the tensile stress pattern in cylindrical organs. (A–C, E–G and I–K) Z-projections (maximal intensity) of confocal stacks from representative, propidium iodide stained, shoot apex (A–C), dark-grown hypocotyl (E–G) and light-grown hypocotyl (I–K), from wild type grown on 1% agar medium (A, E and I), *qua1-1* grown on 1% agar medium (B, F and J) and *qua1-1* grown on 2.5% agar (C, G and K). (D, H and L) Surface signal extracted from confocal stacks, of representative, *GFP-MBD*, shoot apex (D), dark-grown

*Figure 3 continued on next page*

*Figure 3 continued*

hypocotyl (**H**) and light-grown hypocotyl (**L**) grown on 2.5% agar medium. Red lines are the output of the FibrilTool macro, giving a visual representation of the cortical microtubule arrays orientation and anisotropy. The polar histograms in C and K show the global distribution of cell separation orientation in 8 samples from three biological replicates. The polar histograms in D, H and L show the distribution of microtubule array orientations for 12 samples. (**M**) Schematic representation of the relationship between tensile stress and cell separation in the *qua1-1* mutant, and cortical microtubule array organization in the wild type. An horizontal tension will pull the cells apart and create a vertical gap between the cells, perpendicular to the tensile stress pattern, while the microtubules in a wild type context will align horizontally, parallel to the tensile stress pattern. (**N**) Schematic representation of the effect of pressure and growth derived stress on the resulting tensile stress pattern in a cylinder. Shape derived stress is caused by the global turgor pressure of the tissue that puts the epidermis under tension. For a cylindrical shape, this results in a tension twice as high in the transverse direction than in the longitudinal direction (*Beer and Johnston, 1992*). Unidirectional growth driven by the inner tissue layers can also put the outer wall of the epidermis under tension, prescribing maximal tensile stress parallel to the growth direction. Scale bars, 50 μm.

The online version of this article includes the following figure supplement(s) for figure 3:

**Figure supplement 1.** Dark-grown hypocotyl growth and cell separation patterns.

**Figure supplement 2.** Reduction of heterogeneity in pectin esterification in *qua1-1* has no major impact on cell separation patterns.

**Figure supplement 3.** CMT behavior in *GFP-MBD* plants grown on 1% agar.

modulate the matrix potential of the growth medium. We also supplemented the medium with NPA, an inhibitor of polar auxin transport and floral organogenesis, to produce naked meristems that are more amenable to visualization and quantification (see e.g. [*Sassi et al., 2014*]). Strikingly, growth on low water potential medium almost completely restored the formation of normal meristems in *qua1-1* with no or only minor loss of cell adhesion (*Figure 3C*, see also *Figure 1M–P*), thereby allowing us to investigate the gaping pattern in *qua1-1* inflorescence stems.

In *qua1-1* plants grown in normal condition (1% agar), we could not distinguish a clear pattern of cell separation, where in most cases the meristem shape was severely affected and some cells seemed to start proliferating randomly (*Figure 3B*). However, when grown on 2.5% agar medium, longitudinal stripes of bright propidium iodide staining were observed in the *qua1-1* mutant stems; such signal could not be detected in the wild type in the same growth conditions (*Figure 3C*). Some of these stripes developed further into cracks between adjacent cells later on, confirming that the stripes correspond to slight separations between adjacent cells where propidium iodide can accumulate, due to the opening of the cuticle at cell junctions and likely unpacking of the cell wall polysaccharides (*Figure 1—figure supplement 5*). In order to quantify these orientations more precisely, we used our cell separation image analysis pipeline, this time focusing on the orientation of cell separation (see *Figure 1—figure supplement 4*). In the stem apex, we obtained a mean gap angle ($\theta_G$) of 91 ± 7° (*n* = 8 samples), relative to the transverse axis of the stem.

Longitudinal cell separation reveals that the cells are being pulled apart transversely, indicating that maximal epidermal tension is transverse to the axis of the stem. This pattern is consistent with shape-derived stress, assuming that the epidermis acts as a load-bearing layer under tension in that tissue, and thus as in a cylindrical pressure vessel where maximal tension is also transverse (*Figure 3N*).

Note that the age of cell walls may bias our analysis. In particular, based on our results on stem apices, one could propose that older cell walls become less adhesive or more prone to separate. To explore that hypothesis further, we took advantage of our comparative analysis between different tissues to test whether that hypothesis could also hold true.

## Growth-derived stress dominates in hypocotyls

While the stem apex grows relatively slowly, hypocotyls grow fast and primarily in one direction, through anisotropic cell expansion (*Gendreau et al., 1997*). In hypocotyls, cell separations happened in several orientations, leading to epidermal cell naturally peeling out of the surface (*Figure 3*; see also *Figure 1—figure supplement 1*). This pattern is thus not consistent with the pressure vessel model in which stress depends only on shape. To explain this discrepancy, we explore the possible contribution of growth-derived stress: the anisotropic expansion of the inner tissues would pull the load-bearing epidermis longitudinally, thus exerting a longitudinal tensile stress on the epidermis (for predictions of longitudinal stress patterns in growing cylindrical organs, see for example [*Baskin and Jensen, 2013*; *Vandiver and Goriely, 2008*]).

To go beyond this qualitative assessment, we next focused on dark-grown hypocotyls, since they display a well characterized gradient of growth during their elongation, in which cells closest to the root (rootward) have already extensively elongated and are undergoing growth arrest, cells more toward the middle are rapidly elongating, and cells at the top (shootward) and in the apical hook are only starting to elongate (*Figure 3—figure supplement 1A*, [*Gendreau et al., 1997*; *Bastien et al., 2016*]). To ensure phenotypic consistency, we observed the rootward part of dark-grown hypocotyls. Because cracks in *qua1* emerge and develop through growth, we reasoned that the gaping pattern in this part of the hypocotyl would reflect the stress pattern in cells that previously experienced their maximal elongation phase, *a posteriori* (*Figure 3—figure supplement 1A*). When grown on 1% agar, cell separation patterns were so extensive in dark-grown *qua1-1* hypocotyls that the cell separation patterns could not be quantified properly (*Figure 3F*). When grown on 2.5% agar, we could identify discrete cell separations happening almost exclusively transversely to the axis of dark-grown *qua1-1* hypocotyl (*Figure 3G*). Yet, the presence of cells peeling off suggests that longitudinal cell-cell separation also occurs along the longitudinal axis, likely after the initial transverse separations. Interestingly, this is not consistent with the hypothesis that older cell walls separate first, since all epidermal walls in the hypocotyl have the same age after embryogenesis.

To check whether this pattern depends on mechano-chemical polarities in the anticlinal epidermal cell wall, we next overexpressed *PECTIN METHYLESTERASE INHIBITOR 5* (*PMEI5*) in *qua1-1*, reasoning that *PMEI* overexpression should reduce heterogeneity of pectin esterification in the hypocotyl (*Wolf et al., 2012*; *Müller et al., 2013*; *Peaucelle et al., 2015*). Using atomic force microscopy, it was previously shown that wild-type hypocotyls exhibit strong differences in apparent elastic moduli between transverse and longitudinal anticlinal walls in the epidermis, whereas overexpression of *PMEI* significantly reduced such mechanical polarities (*Peaucelle et al., 2015*). As also shown before in *PMEI* overexpressor lines, we observed an increased twisting in the *qua1-1 p35S:: PMEI5* line. Yet, we could not detect significant differences in the gaping pattern between dark-grown *qua1-1* and *qua1-1 p35S::PMEI5* hypocotyls, further confirming that the gaping pattern primarily results from the tension pattern (*Figure 3—figure supplement 2*).

The gaps in dark-grown hypocotyls were too large for our image analysis pipeline to precisely discern cell separation orientation (*Figure 3G*). We thus manually counted the number of events of cell separations that happened either at the transverse or longitudinal junction between adjacent cells. In thirteen images from individual hypocotyls, we counted on average about 11 events of cell separation per image. In total, we found 135 events in which cells had separated along their shared transverse wall, and nine events in which they had separated along their shared longitudinal wall. However, among these nine events, seven were related to an adjacent event of transverse cell separation (*Figure 3—figure supplement 1D and E*), while only two events were strictly longitudinal (*Figure 3—figure supplement 1B and C*). Note that cells at the shootward portion of the hypocotyl, where rapid elongation has not started yet, very few cells were separated in *qua1-1* (*Figure 3—figure supplement 1F*), further supporting the role of anisotropic growth in generating the observed gaps.

To further test whether these gaps can indeed be related to growth-derived stress, we took advantage of the ability of hypocotyls to modulate their growth rate according to light conditions: when grown in light, hypocotyls usually reach 1.5 to 2 millimeters in length, in contrast to dark-grown hypocotyls which can reach about two centimeters (*Gendreau et al., 1997*). We reasoned that, in light conditions, the reduction of elongation should decrease the extent of gap opening in *qua1-1*, while growth anisotropy would still prescribe longitudinal growth-derived stress. As in dark-grown hypocotyls, when grown on 1% agar, cell separation patterns were so extensive in light-grown *qua1-1* hypocotyls that the cell separation patterns could not be quantified properly (*Figure 3J*). When grown on 2.5% agar, transverse gaps were detected in light-grown *qua1-1* hypocotyl (*Figure 3K*). In contrast to dark-grown hypocotyls, we almost only observed slight cell separations in light-grown hypocotyls, as marked by bright propidium iodide staining, consistent with reduced elongation. Because the gaps were much smaller, we could use our pipeline and measured a mean $\theta_G$ of $5 \pm 30°$ (*Figure 3K*; *Figure 1—figure supplement 4*, $n = 8$ samples). A similar response was observed in the *qua1-1 PMEI-OE* line, consistent with the pattern primarily resulting from growth-derived stress, and not from heterogeneities of pectin esterification (*Figure 3—figure supplement 2*).

Altogether, our quantifications support the idea that transverse shape-derived tensile stress dominates in the stem apex epidermis, whereas longitudinal growth-derived stress dominates in the elongating hypocotyl epidermis.

## A mechanical conflict at the petiole-blade junction of the cotyledon

Next we investigated the gaping pattern in *qua1-1* cotyledons where more complex shape and growth patterns occur. In comparison to stems and hypocotyls, cotyledon growth occurs mainly in 2D and is rather isotropic (*Zhang et al., 2011*). Using our pipeline, we analyzed cell separation orientation, focusing on cotyledons at a very young stage (3-day old) in order to observe very early cell separations before gaps become too large. No preferential gap orientations could be detected (*Figure 4C*). Note that while an apparent bimodal distribution (with more separations happening at a 0° and 90° angle), the population of angles could be considered uniformly distributed, as assessed using the Rao's spacing test for uniformity (non-parametric test due to the apparent bimodal distribution of the angle population, and adapted for directional data, p-value=0.5, *n* = 12 samples). This contrasts with the stem apex and the light grown hypocotyl cell separations, for which the test revealed a non-uniform distribution (p-value<0.001 for both). To obtain a different indicator of the spread of the population of angles, we also measured the resultant vector length (*R*) calculated on these populations of angles. *R* lies between 0 and 1; the higher it is the more clustered and unidirectional are the data, while a very low value reveals no preferential orientation. *R* reached 0.84 and 0.70 for the stem apex and light-grown hypocotyls respectively, whereas *R* was equal to 0.07 for cotyledons. Overall these tests reveal that cell separation and thus tensile stress in the cotyledon epidermis is globally isotropic.

When growing seedlings on a 2.5% agar medium, while almost no more cell separation could be observed on the cotyledon blade (see *Figure 1H and J*), the junction between the petiole and the isotropically expanding cotyledon blade exhibited large cell separations (*Figure 4D and I*). The discrepancy in the extent of cell adhesion defects between blade and junction, as revealed by modulating the water potential, suggests that tensile stress is higher at the blade-petiole junction. When measuring the mechanical properties of the petiole-blade junction with AFM, we actually found that this region exhibits increased turgor pressure, apparent stiffness and surface tension (*Figure 4—figure supplement 1*). Interestingly, using the plasma-membrane-associated protein BREVIS RADIX LIKE 2 (BRXL2) as a polar marker, the proximal region of leaves was recently shown to exhibit increased mechanical stress-driven planar cell polarity (*Bringmann and Bergmann, 2017*). This suggests that the predicted mechanical conflict in the petiole-blade junction of the cotyledon is also present in leaves. To check this, we analyzed the gaping pattern in the third leaf in *qua1-1* and found a pattern roughly similar to that of cotyledons, with longitudinal cracks at the base of the leaf (*Figure 4—figure supplement 2*). Interestingly, in the blade of these leaves, we also observed radial cracks around trichomes, a pattern that is also consistent with the recently identified tensile stress pattern in trichome socket cells (*Hervieux et al., 2017*). Furthermore, gaps at the junction appeared preferentially, but not exclusively, along the longitudinal axis of the petiole (*Figure 4D and I*). Our pipeline revealed that cell adhesion defects displayed a mean $\theta_G$ of 85 ± 50° (*Figure 4D and I*, *n* = 8 samples). Note however that the resultant vector length *R* was only equal to 0.21 and the Rao's spacing test showed uniformity of the distribution of angles (*p*-value of 0.5), which suggest that the bias is weak. Altogether, the analysis of the gaping pattern suggests the presence of a mechanical conflict at the petiole-blade junction in cotyledons, both in intensity and direction (*Figure 4I*).

## In the wild type, cortical microtubule orientations match tensile stress patterns, as inferred from cell adhesion defects in *qua1*

To test whether the inferred stress patterns in *qua1-1* are consistent with predicted patterns of stress in the wild type, we next analyzed cortical microtubules (CMTs) in corresponding wild-type organs. CMTs have previously been shown to align along predicted maximal tensile stress direction (*Green and King, 1966*; *Williamson, 1990*). Such response was observed in sunflower hypocotyl (*Hejnowicz et al., 2000*), *Arabidopsis* shoot apical meristems (*Hamant et al., 2008*), leaves (*Jacques et al., 2013*), cotyledons (*Sampathkumar et al., 2014*) and sepals (*Hervieux et al., 2016*). To check whether tensile stress patterns in aerial organs, as inferred from *qua1-1* cracks, match CMT orientations, we next analyzed the CMT behavior in plants expressing the *p35S::GFP-MBD*

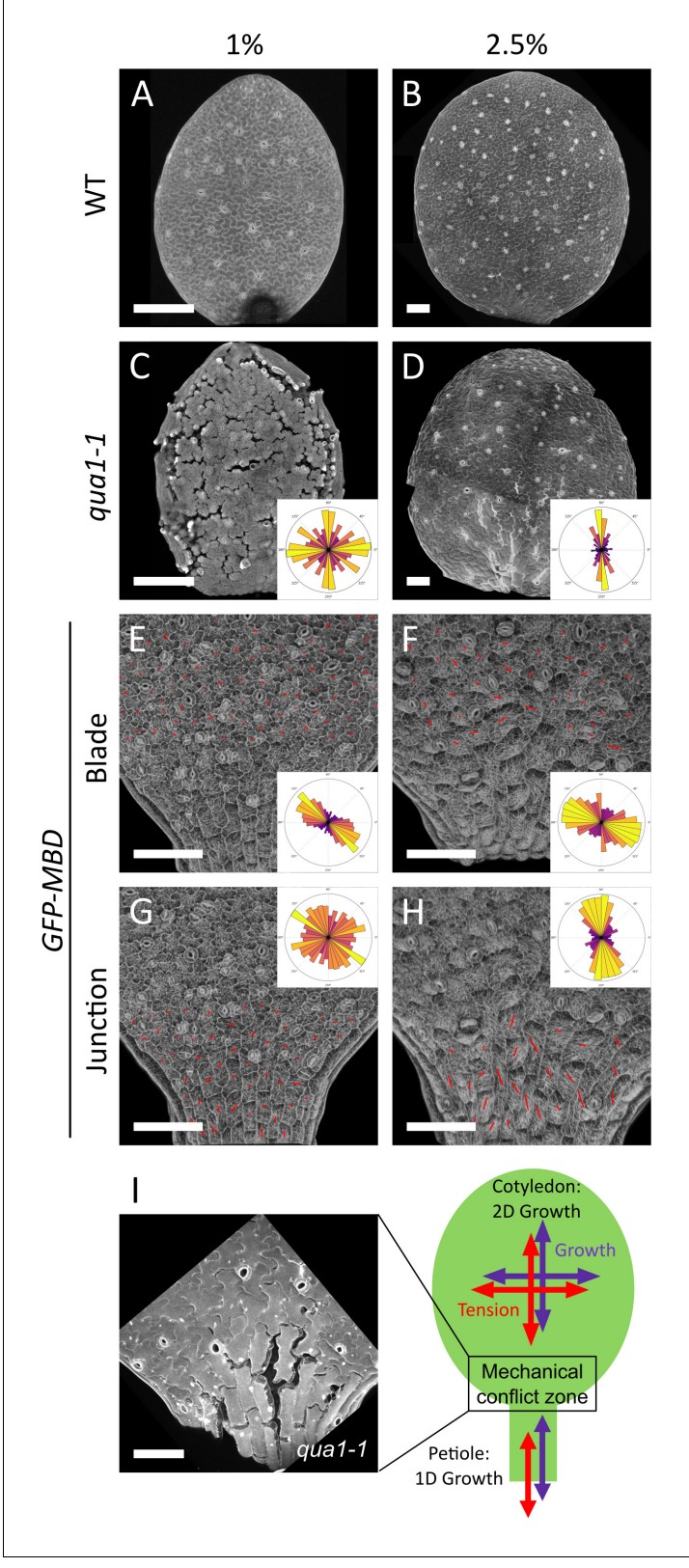

**Figure 4.** A mechanical conflict at the petiole-blade junction in cotyledons. (A–D) Z-projections (maximal intensity) of confocal stacks from representative, propidium iodide stained, 3-day (A) or 5-day (B) old WT cotyledons and 3-day (C) or 5-day (D) old *qua1-1* cotyledons grown on 1% (A, C) and 2.5% (B,D) agar medium. (E–H) Surface signal extracted from confocal stacks, of representative *GFP-MBD* cotyledons grown on 1% (E,G; n = 12 cotyledons) and
*Figure 4 continued on next page*

*Figure 4 continued*

2.5% (F,H; n = 13 cotyledons) agar medium. Red lines are the output of the FibrilTool macro for the blade region (E (n = 898 cells), F (n = 639 cells)) and petiole-blade junction (G (n = 550 cells), H (n = 425 cells)). The polar histograms in C and D show the global distribution of cell separation orientation in *qua1-1* cotyledons (respectively 12 and 8 samples from three biological replicates); the polar histograms in E-H show the distribution of microtubule array orientations for 12 samples. (I) Schematic representation of the effect of differential growth from the cotyledon and the petiole resulting in a mechanical conflict at the junction with higher tensile stress. The close-up image is extracted from a petiole-blade junction in the *qua1-1* mutant. Scale bars, 100 µm.

The online version of this article includes the following figure supplement(s) for figure 4:

**Figure supplement 1.** Mechanical assessment of the cotyledon's petiole-blade junction by AFM.
**Figure supplement 2.** Gaping pattern in the third leaf of *qua1-1* seedlings.

---

microtubule marker (WS-4 ecotype), and quantified the average CMT array orientation ($\theta_M$) and anisotropy using the FibrilTool ImageJ macro (*Figure 5—figure supplement 1*, [*Boudaoud et al., 2014*]).

First, we analyzed CMT orientations in stem apices and cotyledons, where CMT response to mechanical perturbation was already established (*Hamant et al., 2008*; *Sampathkumar et al., 2014*). CMTs at the stem apex were oriented transversely, matching the stress pattern revealed by the *qua1-1* cell separation patterns (*Figure 3D*; for plants grown on 2.5% agar medium: Mean $\theta_M$ = 3 ± 26°, Mean anisotropy = 0.20 ± 0.08, R = 0.65, n = 1669 cells in 13 samples). Note that similar trends were observed on stem apices from seedling grown on 1% and 2.5% agar medium, albeit with a slightly increased noise for seedlings grown on 1% agar medium (*Figure 3—figure supplement 3A,B,E,F*; for plants grown on 1% agar medium: Mean $\theta M$ = 2 ± 26°, Mean anisotropy = 0.21 ± 0.09, R = 0.66, n = 1604 cells in 12 samples).

CMT exhibited different behaviors between the blade and the petiole-blade junction, consistent with the mechanical conflict revealed by the gaps in *qua1-1* (*Figure 4E–H*). When grown on 1% agar medium, CMTs in the blade exhibited a significant bias in their orientation (Mean $\theta_M$ = 150 ± 43°, R = 0.32, n = 898 cells in 12 cotyledons), whereas CMTs at the junction did not exhibit a significant bias in their orientation (R = 0.06, n = 550 cells in 12 cotyledons). Interestingly, the mean anisotropy of the CMT arrays was slightly increased at the junction, when compared to the blade, consistent with the measured increased tension and mechanical conflict in that region (Mean anisotropy = 0.08 ± 0.04 (blade) vs. 0.11 ± 0.06 (junction); paired comparisons between blade and junction from each sample revealed a significant difference (always higher in junction) (Wilcoxon-test, p-values<0.05) for 10 out of 12 samples in each condition). When grown on 2.5% agar medium, the biases in CMT orientation were weak, but significantly different, between the blade and in the junction (blade: mean $\theta_M$ = 154 ± 51°, R = 0.19, n = 639 cells in 13 cotyledons; junction: mean $\theta_M$ = 97 ± 45°, R = 0.28, n = 425 cells in 13 cotyledons). As on 1% agar medium, the anisotropy of the CMT arrays was higher at the junction (0.18 ± 0.09) than in the blade (0.08 ± 0.05). Paired comparisons between blade and junction from each sample revealed a significant difference (always higher in junction) for 12 out of 13 samples in each condition (Wilcoxon-test, p-values<0.001).

In dark-grown hypocotyls, previous studies had reported that CMTs behave differentially at the inner and outer face of the epidermal cells of the hypocotyl and also display different responses at the hypocotyl shootward region (where cells are not elongated yet), middle (where cell elongation rate is high) and rootward region (where cell elongation is slowing down) (*Chan et al., 2010*; *Chan et al., 2011*; *Crowell et al., 2011*). In particular, CMTs were shown to be transversely oriented in the inner face of the epidermal cell at most of these stages, promoting the anisotropic expansion of the tissue (*Crowell et al., 2011*), whereas CMTs display a puzzling behavior at the outer epidermal face: they are rotating in the top and middle part, and are longitudinal in the bottom part. Focusing on the bottom part of the hypocotyl, where cell separation was observed in *qua1-1*, we observed that CMTs were highly aligned longitudinally in the wild type, when grown in the same conditions as in *qua1-1* (*Figure 3H*; seedlings grown on 2.5% agar medium: Mean $\theta_M$ = 98 ± 6°, Mean anisotropy = 0.47 ± 0.06, R = 0.98, n = 182 cells in 12 samples), thus following the tensile stress pattern revealed by the cell separations in *qua1-1*. Note that similar trends were observed for seedling grown on 1% and 2.5% agar medium (*Figure 3—figure supplement 3C,G*; seedling grown

on 1% agar medium: Mean $\theta_M = 93 \pm 6°$, Mean anisotropy = 0.46 ± 0.04, $R$ = 0.97, $n$ = 137 cells in 10 samples).

In light-grown hypocotyls, CMT orientations seemed random at first sight (see e.g. the scatter plot of CMT orientations in *Figure 5—figure supplement 2D* at t = 0 hr). Consistently, CMTs in light grown-hypocotyl have also been described to harbor a rotating behavior (*Chan et al., 2010*). Yet, when taking into account the anisotropy level of the CMT arrays (to put more weight on CMT orientations when anisotropy is high, as done throughout in this article), we revealed a significant bias towards longitudinal CMT orientations, that is parallel to the tensile stress pattern revealed by the cell separations in *qua1-1* (*Figure 3L*, seedling grown on 2.5% agar medium: Mean $\theta_M = 113 \pm 30°$, Mean anisotropy = 0.25 ± 0.10, $R$ = 0.59, $n$ = 606 cells in 11 samples). Note that similar trends were observed for seedling grown on 1% and 2.5% agar medium (*Figure 3—figure supplement 3D,H*; seedling grown on 1% agar medium: Mean $\theta_M = 109 \pm 28°$, Mean anisotropy = 0.21 ± 0.08, $R$ = 0.6, $n$ = 413 cells in seven samples). In comparison to dark-grown hypocotyls, the standard deviation was higher, anisotropy and resultant vector length were lower. Note that anisotropy and resultant vector length are describing different properties: anisotropy relates to CMT arrays in individual cells, while vector length relates to the global behavior of CMT arrays in the cell population. Thus lower anisotropy and vector length in light-grown hypocotyls indicates a global reduction of cellular and supracellular CMT alignment. Altogether, these analyses show CMTs globally follow the stress pattern inferred from the cell separations in *qua1-1* (*Figure 3K and L*).

To further explore the relation between mechanical stress and CMT orientation in the hypocotyl, we next analyzed the CMT response to ablation in this tissue. Assuming that the epidermis is under tension, ablations should disrupt pre-established stress pattern and lead to circumferential tensile stress pattern and CMT orientations (*Hamant et al., 2008*; *Sampathkumar et al., 2014*). As expected, reorientation of CMT arrays was observed in hypocotyls 8 hr after ablation, confirming that CMTs can align along maximal tension in hypocotyls too (*Figure 5D and E*). Interestingly, the CMT orientation was however not fully circumferential after ablation. In particular, we sometimes observed radial CMT orientations at the opposite edges along the longitudinal axis of the ablation site, matching the growth-derived longitudinal tensile stress pattern in the hypocotyl (*Figure 5—figure supplement 2A and B*). Conflicts between superposing mechanical stress patterns have been modeled and reported before, notably at the organ-boundary in shoot apical meristems: in that domain, the circumferential CMT orientation after ablation is also mixed, because the boundary is a site of highly anisotropic stresses (see Figure S7 in [*Hamant et al., 2008*]). If such a conflict was not present, one would expect a true circumferential CMT orientation (*Figure 5—figure supplement 2C*). In contrast, our quantifications of the mixed CMT orientations after ablation implies that such a conflict of stress patterns is prominent in hypocotyls (*Figure 5—figure supplement 2D–F*), further consolidating the presence of longitudinal tensile stress in the hypocotyl epidermis.

Overall, CMT orientation and anisotropy correlate well with the tensile stress pattern inferred from the cell separation patterns in *qua1-1*.

## Epidermis continuity is required for tensile stress propagation and the supracellular alignment of cortical microtubules

In plants, the outer epidermal wall is inherited from the zygote and its continuity may be one of the key element to propagate tensile stress over several cell files (*Galletti et al., 2016*). Yet, this still remains to be demonstrated. The analysis of CMTs in *qua1-1* offers us the unique opportunity to test that hypothesis. We thus investigated whether the supracellular microtubule alignment with maximal tensile stress requires an intact outer wall.

We introgressed the *GFP-MBD* microtubule reporter in the *qua1-1* mutant and analyzed microtubule behavior in light-grown hypocotyls, where cracks are mostly superficial early on. Whereas CMTs looked normal within individual *qua1-1* cells, they displayed less consistent orientation at the tissue scale in *qua1-1* cells than in the wild type (*Figures 3L* and *5A*). When grown on 1% agar medium, for *GFP-MBD* and *qua1-1 GFP-MBD,* the mean $\theta_M$ was 109 ± 28° and 107 ± 40°, the mean anisotropy was 0.21 ± 0.08 and 0.21 ± 0.09 and $R$ was 0.6 and 0.36, respectively ($n$ = 413 cells in 7 samples and 1019 cells in 16 samples respectively). In other words, the distribution of CMT angles was broader in the *qua1-1* mutant. Note that this discrepancy was also registered when calculating the CMT orientation resultant vector length for each plant individually (allowing us to compare cell populations and test whether the difference is significant): we found a mean resultant vector length for *GFP-MBD*

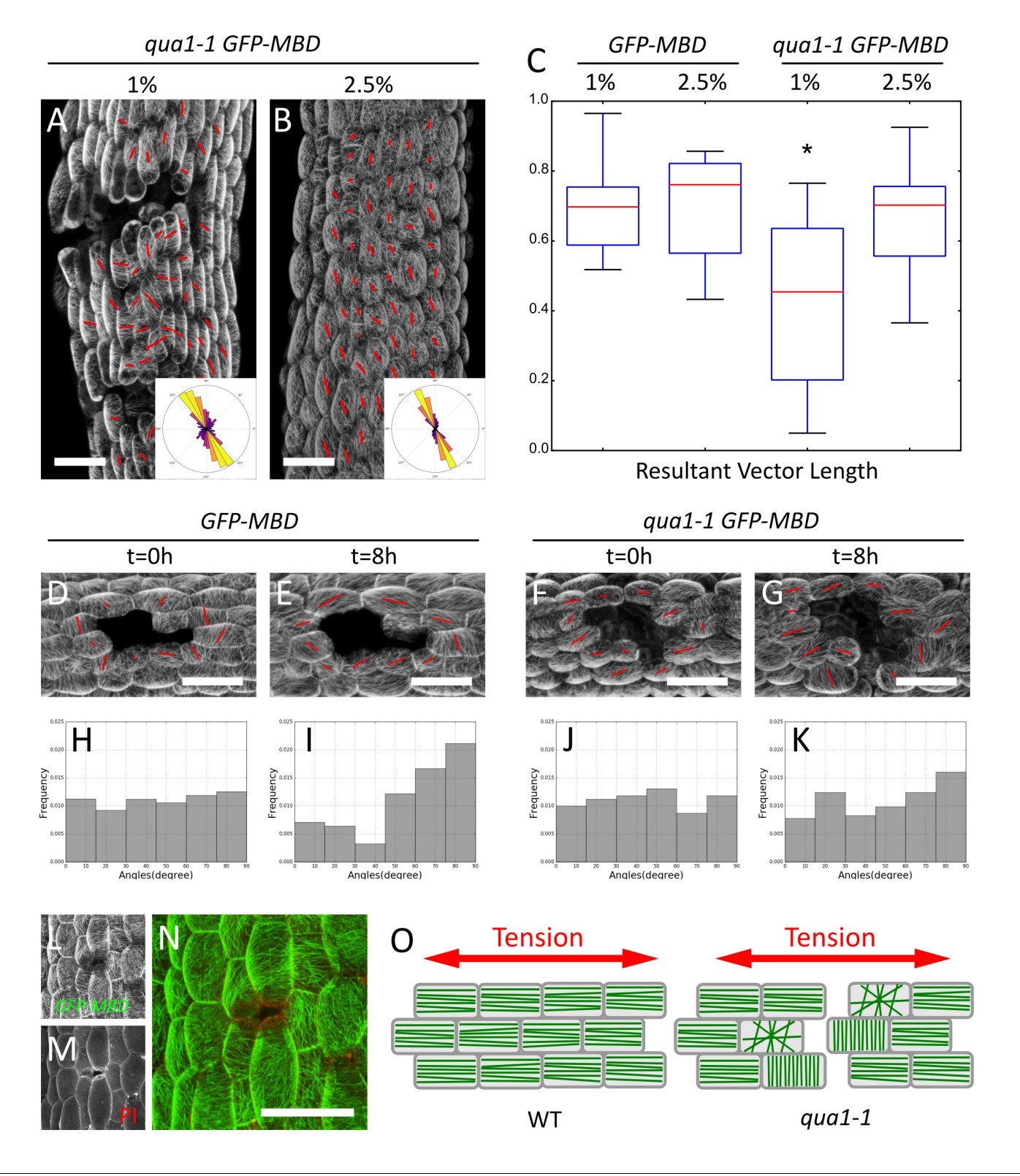

**Figure 5.** Epidermis continuity is required for tensile stress propagation and supracellular alignment of cortical microtubules. (A,B) Surface signal extracted from confocal stacks of representative *qua1-1 GFP-MBD* light-grown hypocotyls. Red lines in A and B are the output of the FibrilTool macro. Cell separation affects cortical microtubule arrays cell to cell coordination across the tissue as compared to *Figure 3L*. The polar histograms in A and B show the global distribution of microtubule array orientations in 15 (A) and 16 (B) samples from three biological replicates. (C) Box plot of the resultant

*Figure 5 continued on next page*

*Figure 5 continued*

vector length revealing less consistent CMT alignment in *qua1-1* hypocotyls grown on 1% agar medium, and a rescue of this defect in *qua1-1* hypocotyls grown on 2.5% agar medium. (D–K) Ablation experiment showing the reorganisation of CMT in response to the modified stress pattern in *GFP-MBD* (D and E) and *qua1-1 GFP-MBD* (F and G), at t = 0 hr after the ablation (D and F) and t = 8 hr after the ablation (E and G) (10 samples from three biological replicates for each condition). (H–K) Histograms of the distributions of CMT array angles relative to the ablation site, and corresponding to panels D-G. (L–N) Local impact of a cell-cell adhesion defect in *qua1-1 GFP-MBD*: CMTs align along the local maximum of tension around the gap. (L) *GFP-MBD* signal, (M) Propidium Iodide signal, (N) overlay. (O) Schematic representation of the effect of cell separation in the *qua1-1* mutant on cortical microtubule array organization. In the wild type, tension is propagated across the tissue in the epidermis, leading to consistent CMT array alignments in the tissue. In contrast, in *qua1*, the mechanical discontinuity of the epidermis hinders the normal propagation of the tension across the tissue and locally affects the pattern of stress, leading to less pronounced CMT array alignments in the tissue. Scale bars, 50 µm.

The online version of this article includes the following figure supplement(s) for figure 5:

**Figure supplement 1.** Surface signal extraction and semi-automated CMT array quantification.

**Figure supplement 2.** Conflict between ablation-derived (circumferential) and growth-derived (longitudinal) tensile stress in the hypocotyl outer epidermal cell wall.

---

samples of 0.69 ± 0.14 and 0.42 ± 0.23 for *qua1-1 GFP-MBD* and the populations were significantly different (*t*-test p-value=0.012).

Our analysis quantifying the coordination of CMTs across the whole tissue shows that cells in *qua1-1* hypocotyls are less coordinated across the tissue than in the wild type. The presence of cracks may be the main reason for this response: cracks generate local perturbations in the stress pattern, such that CMTs tend to reorganize around these cracks, as shown for cell ablations (*Figure 5L–N* and e.g. [*Hamant et al., 2008*; *Hervieux et al., 2016*]). In turn, the actual cell-to-cell coordination does not seem to be affected when two neighboring cells are still attached. Consistently, the anisotropy of CMTs in *qua1-1 GFP-MBD* was not decreased compared to the wild type. In many instances the anisotropy of CMT arrays was higher in *qua1-1 GFP-MBD* and CMTs were somehow oriented following the contours of large gaps, thus apparently following a new tension pattern allowed by the continuity of the cells where they are still adhesive (*Figure 5A,L–N*). Our analysis thus reveals that local cell separations lead to the loss of the tissue-scale organization of the CMTs that is usually found in the wild-type. Therefore, while these data further support a scenario in which epidermis continuity is required for supracellular CMT behavior, the cell-cell separations are too strong to conclude.

When grown on 2.5% agar medium, the wild type and *qua1-1* mutant hypocotyls exhibited similar CMT behaviors: we found a mean resultant vector length for *GFP-MBD* (2.5% agar) samples of 0.69 ± 0.15 and 0.65 ± 0.17 for *qua1-1 GFP-MBD* (2.5% agar) (n = 606 cells in 11 samples and 865 cells in 15 samples, respectively) and the populations were not significantly different (*t*-test p-value=0.556). This is consistent with the phenotypic rescue of the cracks in these conditions, which likely restores the mechanical continuity of the epidermis (*Figure 5B,C*). To test the contribution of epidermis integrity for supracellular CMT orientation along maximal tensile stress, and building on the observed rescue of CMT behavior on 2.5% agar medium, we next modified the mechanical stress pattern in *qua1-1* artificially, using growth conditions in which cracks are only starting to appear. As already shown, cortical microtubule orientation became mostly circumferential in wild type hypocotyls 8 hr after ablation (*Figure 5D and E*). Strikingly, this response was dramatically reduced in *qua1-1* (*Figure 5F and G*). To quantify this response, we calculated the acute angle between the ablation site and the main orientation of the CMT arrays obtained with the FibrilTool macro, for each cell, at t = 0 hr and t = 8 hr after the ablation (*Figure 5—figure supplement 2K*). We thus obtained angles ranging from 0° to 90° (90° corresponding to circumferential CMT orientations around the ablation site, that is parallel to predicted maximal tension). At t = 0 hr, the *GFP-MBD* line exhibited a mean angle of 47 ± 26° showing an homogeneous distribution and no preferred orientation of the microtubules relative to the ablation site (n = 101 cells in 10 samples; *Figure 5H*). At t = 8 hr the mean angle shifted to 58 ± 25°, and the distribution exhibited a strong skewing towards the 80 to 90° angles (n = 101 cells in 10 samples; *Figure 5I*), demonstrating a significant reorganization of the CMT arrays around the ablation site (p-value=0.002). The *qua1-1 GFP-MBD* line however, did not show a significant reorganization of its CMT arrays 8 hr after the ablation (*Figure 5J and K*, Mean angle at t = 0h: 45 ± 26° and t = 8h: 50 ± 26°, n = 107 cells in 10 samples; p-value=0.186). Whereas the distribution of angles still showed a relative skewing towards 90°, the

skewness was much smaller than in the *GFP-MBD* line: −0.23 in *qua1-1 GFP-MBD* t = 8 hr vs. −0.77 in *GFP-MBD* t = 8 hr (0,08 in *qua1-1 GFP-MBD* t = 0 hr and −0.04 in *GFP-MBD* t = 0 hr, *Figure 5K*). In addition a statistical test comparing the skewness of the distribution to a corresponding normal distribution showed that the population of angles are not significantly skewed in *GFP-MBD* t = 0 hr, *qua1-1 GFP-MBD* t = 0 hr and *qua1-1 GFP-MBD* t = 8 hr (p-value=0.858, 0.718 and 0.255 respectively), while *GFP-MBD* t = 8 hr shows a significant skewing (p-value=0.002).

Overall, these results demonstrate that a discontinuous outer wall hampers the ability of CMTs to align with supracellular maximal tension in the epidermis (*Figure 5O*).

## Discussion

A defining feature of the epidermis in animals is its continuity, which allows a build-up of tension and in turn promotes adhesion and the coordinated behavior of epidermal cells, notably through local cadherin-based cell-cell adhesion (*Galletti et al., 2016*). This loop can be referred as the 'tension-adhesion feedback loop'. The conclusions from the present study support this picture in plants too, notably as tension patterns could be revealed by the early cell-cell adhesion defects in *qua1-1* and as adhesion defects in *qua1-1* hinder the propagation of tensile stress (*Figure 6*).

The physics of pressure vessel, and more generally of solid mechanics, has been instrumental in the derivation of stress patterns in plants, very much in the spirit of D'Arcy Thompson's legacy (*Thompson, 1917*). Yet, the role of mechanical stress in cell and developmental biology remains a subject of heated debate, notably because forces are invisible and stress patterns can only be predicted. Here we used the simplest possible tool to reveal stress pattern in plants, building on cell adhesion defects and their relation to tension. Our results formally validate several conclusions from previous computational models and stress patterns deduced from ablation experiments. Incidentally, this work also brings further experimental proof of the presence of tissue tension, building on the initial work by Hofmeister (*Hofmeister, 1859*) and Sachs (*Sachs, 1878*). We believe this work can serve as a reference for further studies on cell behavior in the corresponding tissues, notably to relate the behavior of cell effectors to stress patterns. Based on these data, one can also revisit previously published work on microtubule behavior in various organs. For instance, it had been reported that CMTs

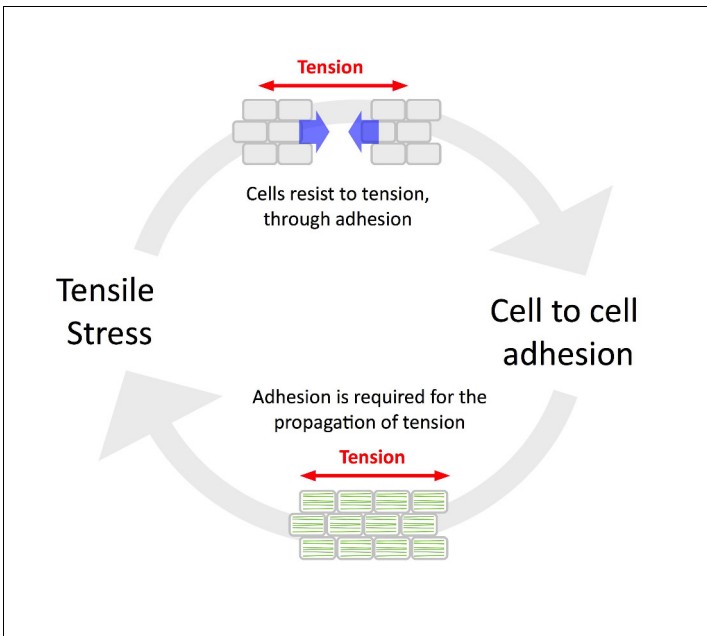

**Figure 6.** The tension-adhesion feedback loop in plant epidermis. Schematic representation of the relationship between tensile stress and cell to cell adhesion in the epidermis. Tensile stress pulls the cells apart in the *qua1* mutant, while cell to cell adhesion is required to allow the normal propagation of tensile stress in the epidermis, as revealed by the organization of CMT arrays in the *qua1* mutants and the wild type. This paradoxical relationship highlights the importance of epidermis continuity for mechanoperception at the tissue scale.

constantly rotate in young hypocotyl until they reach either their transverse or longitudinal orientation (*Chan et al., 2007*). Although this behavior is still an intriguing phenomenon, it could partly be related to the fact that in this tissue, (transverse) shape-derived and (longitudinal) growth-derived stresses are in competition. Slight oscillations in growth rate (*Bastien et al., 2016*) could periodically affect the ratio of shape- and growth-derived stress and thus trigger this rotating behavior of the microtubules. Another exciting avenue for the future is the analysis of the integration between such tensile stress patterns and other cues such as blue light, and its effect through phototropin and katanin (*Lindeboom et al., 2013*), or hormones (*Vineyard et al., 2013*) on microtubule reorientation.

By definition, mechanical forces can be propagated at the speed of sound, like acoustic waves. This may provide close to instantaneous coordinating cues to tissues, as proposed in the Drosophila wing disc where cell division stops synchronously (*Shraiman, 2005*). Such a mechanism, in turn requires a tightly maintained and controlled cell-to-cell adhesion (*Verger et al., 2016*) in order to constantly maintain the mechanical conductivity of the tissue. Yet, the heterogeneous nature of the tissue would add noise to the signal propagation, and so far this question has not been formally addressed in developmental biology. Here we provide evidence that the continuity of the outer cell wall is required for the coordinated response of adjacent cells to mechanical stress. In other words, a cell would experience different tension level and orientation between a situation where it would be separated from its neighbors (typically, cell shape would be sufficient to prescribe a tensile stress pattern, as in [*Sampathkumar et al., 2014*]) and a situation where it would still adhere to its neighbors (tensile stress could build up at the outer epidermal wall, to a magnitude higher than cell shape-derived stress, and with a pattern that would depend on tissue shape and differential growth, as in [*Hamant et al., 2008*; *Louveaux et al., 2016*]). This opens exciting prospect for the future, as cell variability and growth heterogeneity is attracting increasing attention because of its, sometimes counterintuitive, instructive function in development (*Hong et al., 2018*).

Note that although CMT alignments match maximal tensile stress direction, as predicted from adhesion defects in *qua1*, this does not necessarily mean that microtubule, and cellulose, become aligned to promote cell-cell adhesion. In fact, mutants in microtubule dynamics or cellulose deposition have not been reported to exhibit adhesion defects. It is therefore more likely that other factors, and most likely actin through its impact on pectin delivery to cell walls (*Mathur et al., 2003*), plays a critical role in the tension-induced reinforcement of cell-cell adhesion, in parallel to microtubule-driven cell wall reinforcement.

Our conclusions also echo the recent analysis of the DEFECTIVE KERNEL 1 (DEK1) protein, which was shown to be required for tensile stress perception *via* its association with a mechanosensitive Ca$^{2+}$ channel (*Tran et al., 2017*). Interestingly, *DEK1* RNA interference lines exhibit loss of cell adhesion in their epidermis (*Johnson et al., 2005*). Our finding in *qua1* thus allows us to revisit these results, suggesting that plant epidermis requires tensile stress perception, and in turn, that cell-cell adhesion allows tensile stress propagation.

While the coordinating role of the outer wall is difficult to match with a comparable structure in animals, the basement membrane may in principle have a similar role, given its continuity and key role in both adhesion and mechanotransduction. Based on our results in plants, the analysis of basement membrane continuity, and its disruption, may very well help understand how consistent supracellular epidermal patterns relate to mechanical stress, in parallel to the well-established role of cadherin and stress in cell-cell adhesion and epidermal functions (*Galletti et al., 2016*).

## Materials and methods

### Plant material, genotyping and growth conditions

The *qua1-1* (WS-4) T-DNA insertion line, the *GFP-MBD* (WS-4) microtubule reporter line the *p35S:: PMEI5* (Col-0) and the *pPDF1::mCit:KA1* (Col-0) L1 expressed plasma membrane marker, were previously reported in (*Bouton et al., 2002*; *Marc et al., 1998*; *Wolf et al., 2012*; *Simon et al., 2016*) respectively. The *qua1-1* line was genotyped by PCR using the primers described in (*Bouton et al., 2002*) and the *p35S::PMEI5* homozygous lines were selected based on their strong phenotype (*Wolf et al., 2012*; *Müller et al., 2013*).

*Arabidopsis thaliana* seeds were cold treated for 48 hr to synchronize germination. Plants were then grown in a phytotron at 20°C, in a 16 hr light/8 hr dark cycle on solid custom-made Duchefa

'Arabidopsis' medium (DU0742.0025, Duchefa Biochemie). Seedling age was counted from the start of light exposure.

For dark-grown etiolated hypocotyls, seeds were exposed to light for 4 hr to induce germination. The plates were then wrapped in three layers of aluminum foil to ensure skotomorphogenesis. Naked shoot apical meristems were obtained by adding 10 µM of NPA (N-(1-naphthyl) phthalamic acid) in the medium as described in (*Hamant et al., 2014*).

For time-lapse images of cell separation dynamics, seedlings were first grown on 'Arabidopsis' medium containing 2.5% agar; once cotyledons just opened, they were mounted on 'Arabidopsis' medium containing 1% agar and imaged for up to 72 hr every 12 hr. During image acquisition, the seedlings were immersed in water supplemented with 1 ml of PPM (PPM-Plant Preservative Mixture, Kalys) per liter of medium to prevent contamination. After each acquisition the water was removed and the plants were placed back in a phytotron (see growth conditions above).

## Low water potential treatments

Water potential of the medium was changed using either higher agar concentration ((1% and 2.5%) [*Owens and Wozniak, 1991*]) or increasing Polyethylene Glycol. We used a PEG-infused plates method adapted from (*van der Weele et al., 2000*). Classic 1% agar 'Arabidopsis' medium (as described above), as well as liquid 'Arabidopsis' medium containing various concentrations of PEG (PEG20000, Sigma-Aldrich) were prepared. The liquid medium was supplemented with 1 ml of PPM (PPM-Plant Preservative Mixture, Kalys) per liter of medium to prevent contamination. The liquid medium osmolarity was measured using a cryoscopic osmometer (Osmomat 030, Gonotec). Solid medium petri dishes were made, let to solidify for about 2 hr, and an equal volume of liquid medium was poored on top. After 24 hr of diffusion, the liquid medium was recovered and its osmolarity was measured again. The increase of osmolarity due to PEG diffusion in the solid medium was deduced from the difference of osmolarity of the liquid medium before and after diffusion. The petri dishes were let to dry for about 2 hr and the seeds were sown.

## Cell wall staining, ablations, confocal microscopy and stereomicroscopy

For cell wall staining, plants were immersed in 0.2 mg/ml propidium iodide (PI, Sigma-Aldrich) for 10 min and washed with water prior to imaging. Ablations were performed as previously described in *Uyttewaal et al., 2012*: seedlings were mounted horizontally with 2% low melting agarose (Sigma-Aldrich, St. Louis, MO, USA) on 'Arabidopsis' medium containing 1% agar and imaged immediately after the ablation and 8 hr after the ablation. The ablations were performed manually with a fine needle (Minutien pin, 0.15 mm rod diameter, 0.02 mm tip width, RS-6083–15, Roboz Surgical Instrument Co.) ablating approximately five epidermal cells and some cells from the inner layers. Because the size of the ablation can vary from one sample to another, ablations were originally performed on a large number of samples and hypocotyls with comparable number of ablated epidermal cells (approx. 5) were further imaged and analyzed. For imaging, samples were either placed on a solid agar medium and immersed in water, or placed between glass slide and coverslip separated by 400 µm spacers to prevent tissue crushing. Images were acquired using a Leica TCS SP8 confocal microscope. PI excitation was performed using a 552 nm solid-state laser and fluorescence was detected at 600–650 nm. GFP excitation was performed using a 488 nm solid-state laser and fluorescence was detected at 495–535 nm. mCitrine excitation was performed using a 514 nm solid-state laser and fluorescence was detected at 520–555 nm. Stacks of 1024 × 1024 pixels optical section were generated with a Z interval of 0.5 µm for *GFP-MBD,* 1 µm for PI or 0.5 µm when *GFP-MBD* and PI were acquired at the same time and 0.25 µm for *mCit:KA1*. Stereomicroscopy images were taken using a leica MZ12 stereo microscope with an axiocam ICc5 Zeiss CCD camera.

## Atomic Force Microscopy

AFM determination of apparent stiffness *k* and turgor pressure *P* in cotyledon epidermis was performed as in (*Beauzamy et al., 2015*) with modifications. Specifically, the adaxial surface of 3 day old cotyledons was measured. Dissected cotyledons grown on different agar concentrations were mounted with Patafix (UHU) and subsequently submerged in water for measurement, whereas whole seedlings grown on different PEG concentrations were mounted in 2% low-melting agarose (Sigma-Aldrich) and submerged in liquid Arabidopsis medium (DU0742.0025, Duchefa Biochemie)

supplemented with D-Mannitol (Sigma-Aldrich) to reach the same osmotic pressure of PEG-infused solid medium. Mannitol was used to prevent potential interference of the high viscosity of PEG solutions, and each measurement was performed under 20 min to reduce osmolite uptake. For additional AFM measurements (*Figure 2—figure supplement 1* and *Figure 4—figure supplement 1*), whole seedlings grown on different agar concentrations were mounted in 2% low melting agarose (Sigma-Aldrich) and immersed in liquid Arabidopsis medium or water depending on the experiment.

A BioScope Catalyst AFM (Bruker) was used for measurement with spherical-tipped AFM cantilevers of 400 nm tip radius and 42 N/m spring constant (SD-SPHERE-NCH-S-10, Nanosensors). For topography, peak force error and DMT modulus images, PeakForce QNM mode of the acquisition software was used, with peak force frequency at 0.25 kHz and peak force set-point at 1 μN for wild-type and 200 nN for *qua1-1* due to their innate difference in stiffness. Larger peak force set-point frequently damaged *qua1-1* sample surface. 128*128 pixels images of 30*30 μm$^2$ area were recorded at 0.1 Hz scan rate. For Young's modulus, apparent stiffness and turgor pressure measurements, 1 to 2 μm-deep indentations were performed along the topological skeletons of epidermal cells to ensure relative normal contact between the probe and sample surface. At least three indentation positions were chosen for each cell, with each position consecutively indented three times, making at least nine indentation force curves per cell. Cell registration of AFM force curves were performed with the NanoIndentation plugin for ImageJ (https://fiji.sc/) as described in (*Mirabet et al., 2018*).

Parameters for turgor deduction were generated as follows. Cell wall elastic modulus *E* and apparent stiffness *k* were calculated from each force curve following (*Beauzamy et al., 2015*). Cell surface curvature was estimated from AFM topographic images, with the curvature radii fitted to the long and short axes of smaller cells or along and perpendicular to the most prominent topological skeleton of heavily serrated pavement cells. Turgor pressure was further deduced from each force curve (100 iterations) with the simplified hypothesis that the surface periclinal cell walls of leaf epidermis has constant thickness (200 nm), and cell-specific turgor pressure is retrieved by averaging all turgor deductions per cell.

AFM-measured mechanical properties were used to deduce outer cell wall tension in cotyledon epidermis. Cells were considered as spherical thin-walled pressure vessels, with the stress equals to

$$\sigma = \frac{Pr}{2t}$$

where *P* is turgor pressure, *r* is the vessel radius as the inverse of cell surface mean curvature, and *t* is assumed cell wall thickness. Since *t* is assumed constant, σ only depends on turgor pressure and surface topography.

## Automated detection of cell separations

The procedure to perform the automated detection of cell separations is described in more detail at Bio-protocol (*Verger et al., 2018*). We developed a semi-automated image analysis pipeline in python language in order to detect and analyze cell separations in a tissue (*Verger and Cerutti, 2018*; copy archived at https://github.com/elifesciences-publications/Cell_separation_analysis). Input images are 2D Z-projections from confocal Z-stack. The script works by segmenting cell separation based on a threshold detection method of pixels intensity. In the case of clear gaps between the cells (as in cotyledons) the pixel intensity is much lower. The threshold allows a segmentation of theses gaps with low intensity pixels. In the case of bright stripes appearing between the cells (as in stem apices or light-grown hypocotyls), an opposite threshold is used, segmenting only the high intensity pixels zones. Because this threshold may vary from one image to the other, it was manually defined using the ImageJ threshold tool before running the script. Running the script then, labels the different zones, measures their areas, performs a principal component analysis of the label in order to determine their main orientations (principal component, $\theta_G$) and assigns an anisotropy and principal angle to each labeled region. The output of the script is an image of every cell separation segmented with a visual representation of their anisotropy, and a polar histogram giving a visual representation of the global result. The python script was developed and run in the TissueLab environment of the OpenAleaLab platform ([*Cerutti et al., 2017*], https://github.com/VirtualPlants/tissuelab) and using functions from the python libraries SciPy (www.scipy.org), NumPy (www.numpy.

org), Pandas (pandas.pydata.org) and Matplotlib (matplotlib.org, see *Figure 1—figure supplement 4*).

## CMT analysis

Original images were confocal Z-stacks. We used the MorphoGraphX software (*Barbier de Reuille et al., 2015*) to recover only the outer epidermal cortical microtubules signal. Further analysis was performed in 2D using the imageJ morpholibJ library (*Legland et al., 2016*), to segment cells, the analyze particle tool to define ROIs and an automated version the FibrilTool macro (*Boudaoud et al., 2014*) to analyze the principal orientation ($\theta_M$) and anisotropy of the CMT arrays (see *Figure 5—figure supplement 1*).

For the CMT response to ablation, only the cells directly surrounding the ablation were analyzed. Points were manually placed and used to calculate the acute angle between the ablation site and the orientation of the microtubule array for each cell (see *Figure 5—figure supplement 1*).

## Statistical analysis

For linear data (comparing gap area, anisotropy), classical statistical tests were used. Normality of the samples was tested using Shapiro's test. If at least one of the sample population did not have a normal distribution, the populations were compared with the non parametric Wilcoxon Rank Sum test. If both samples had normal distributions, their variance were compared using Bartlett's test. If they were equal, a Student's *t*-test was performed; if they were unequal, a Welch's *t*-test was performed.

For circular (or directional) data (cell separation orientation and CMT orientation), different statistics were used. Since in our case a 0° angle is equal to an 180° angle within a semicircle, the mean, standard deviation, variance, and the accompanied statistical test have to take this fact into account and circular statistics were used. Most of our sample did not show a Von Mises distribution (equivalent of the normal distribution), thus only non-parametric tests were used. The Rao's spacing test was used to determine if the populations of angles were homogeneously distributed, or had a preferred orientation determined by the circular mean.

Finally the data form the CMT response to ablation were treated as linear data, since the calculated angle was between 0° and 90°. The symmetry of CMT response distribution was measured by the skewness of the population towards 90° and the significance of the skewness was tested against a normal distribution.

All statistical analyses were performed in python using the scipy.stats library (scipy.org) for linear data and the pycircstat library (github.com/circstat/pycircstat) for circular data.

## Acknowledgements

We thank G Cerutti for help and advices in the design of the cell separation image analysis pipeline, M Liu for technical help, M Dumond for help and advice with statistical analysis. We are thankful to Gwyneth Ingram and Jan Traas for their comments and feedback on this manuscript. This work was supported by the European Research Council (ERC-2013-CoG-615739 'MechanoDevo' and ERC-2012-StG-307387 'Phymorph') and the European Molecular Biology Organization (EMBO ALTF 168–2015).

## Additional information

### Funding

| Funder | Grant reference number | Author |
| --- | --- | --- |
| H2020 European Research Council | ERC-2013-CoG-615739 | Olivier Hamant |
| European Molecular Biology Organization | EMBO ALTF 168-2015 | Yuchen Long |
| H2020 European Research Council | ERC-2012-StG-307387 | Arezki Boudaoud |

The funders had no role in study design, data collection and interpretation, or the decision to submit the work for publication.

## Author contributions
Stéphane Verger, Conceptualization, Data curation, Formal analysis, Investigation, Visualization, Methodology, Writing—original draft, Writing—review and editing; Yuchen Long, Data curation, Formal analysis, Investigation, Methodology, Writing—review and editing; Arezki Boudaoud, Supervision, Writing—review and editing; Olivier Hamant, Conceptualization, Supervision, Funding acquisition, Writing—original draft, Project administration, Writing—review and editing

## Author ORCIDs
Stéphane Verger [ID] https://orcid.org/0000-0003-3643-3978
Olivier Hamant [ID] http://orcid.org/0000-0001-6906-6620

## Decision letter and Author response
Decision letter https://doi.org/10.7554/eLife.34460.sa1
Author response https://doi.org/10.7554/eLife.34460.sa2

# Additional files

## Supplementary files
• Transparent reporting form

## Data availability
All data generated or analysed during this study are included in the manuscript and supporting files. Source data file for cell_separation_analysis pipeline has been provided and a reference to Github (where the code is now stored) has been added in the main text and Materials and methods.

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
