## [Decision Letter]

Thank you for submitting your article "The tension-adhesion nexus in plant epidermis" for consideration by *eLife*. Your article has been favorably evaluated by Christian Hardtke (Senior Editor) and three reviewers, one of whom is a member of our Board of Reviewing Editors.

The reviewers have discussed the reviews with one another and the Reviewing Editor has drafted this decision to help you prepare a revised submission.

Summary:

Probing the role of mechanics in biology is an increasingly popular topic. Here, the authors use a simple but clever method to monitor tension in plants by weakening the connections between plant cell walls. The authors make nice use of the *qua1* mutant, which affects cell adhesion due to defects in the middle lamella, to dissect the connection between cell adhesion, mechanical stress and coupling of stress between cells in a tissue. This system is then followed up by biophysical probes (AFM) and the analysis of MT patterns, as well as imaging and modeling. The ability to deduce stress patterns using this mutant and osmotic manipulations is a significant contribution and the finding that supracellular microtubule alignment is aberrant when cell continuity is disrupted is interesting.

The three reviewers thought the work was well written and presented and that it would, pending some additional small experiments and rewriting, be suitable for *eLife*. The major issues to be addressed are listed below and should be explained in a rebuttal.

Essential revisions:

1) The explanation for "reasoning that PMEI overexpression should induce more homogeneous wall properties in the hypocotyl" need to be explained more clearly and consistently and the overall rationale for this experiment explained. Also, in the conclusion from this experiment, couldn't this negative result just mean the experiment didn't work? Was the change in wall properties measured, or was there some other evidence that *p35S::PME15* was working (some other phenotype that did occur?) The authors should discuss potential caveats of these overexpression experiments such as compensatory changes in wall composition/mechanics. Is *p35S::PMEI5* the same as *PMEI-OE*?

2) In numerous places, experiments were not done on true qual/control pairs under the same conditions, or in same tissues. Below is a (non-exhaustive) list of places where the discrepancies should be addressed.

a) In the last paragraph of the subsection “A mechanical conflict at the petiole-blade junction of the cotyledon”, the authors mention a study done in leaves that they say is consistent with their results in cotyledons. It seems like a relatively straightforward experiment (given their current resources) to include an image of a leaf from qua-1 2.5% agar grown plants and to track the "cracks" using their pipeline to actually address this issue.

b) It is possible, perhaps even likely, that cell wall composition/structure is different between the blade and petiole cells. These differences might also contribute to the observed stress gradient at the blade-petiole junction. The authors should perform AFM analyses in this area to determine whether the cell stiffness and tension differ between petiole cells and their neighboring blade cells.

c) In Figure 2 it is unclear whether wild-type or *qua1* mutants used for AFM analysis – both should be reported.

d) In Figure 2 different osmotic conditions used between experiment (2.5% agar) and AFM (water) measurements may cause alterations in turgor pressure because cells respond in minutes (Shabala and Lew, 2002). Does water immersion interfere with measurements?

e) In Figure 2 use of different osmotic stabilizers between experiment (PEG) and AFM (mannitol) is also problematic. It would be more consistent to perform *qua1* experiments in mannitol if the PEG is too viscous for AFM.

f) In Figure 2 different cell types (cotyledon pavement cell) in AFM versus experimental material (hypocotyl, early cotelydon, petiole/leaf junction).

g) Stiffness was measured on epidermal cells, not on the connections between cells, which are altered in the *qua1* mutant. Does this reduction in stiffness reflect slower growth caused by growing in hyperosmotic conditions?

h) Analysis of CMT organization (at the very top of the apex) in a region that does not match the longitudinal cell wall separation (lower down on the shank of the apex) (compare Figure 3C and 3D). Please perform CMT organization in the relevant part of the apex.

i) Figure 3D compared microtubule organization in wild-type cells expressing GFP-MBD (in 1% agar?) to Figure 3C*qua1* mutants in 2.5% agar. The agar concentration needs to be specified and also must be the same to compare the samples. Please perform your analysis of GFP-MBD in 2.5% agar or, if that was what was done, indicate that in the figure and legend.

3) More experimental detail should be presented for the following (some overlap with items in #2).

a) In the fifth paragraph of the subsection “In the wild type, cortical microtubule orientations match tensile stress patterns, as inferred from cell adhesion defects in *qua1*” please briefly describe the ablation in more detail here in the text and also the methods (the only mention is a needle and a citation). How many cells are ablated? Is it a clean break? Is it only the epidermis? Can the tissue be imaged before this ablation, or only after?

b) Is there a differential response between the mutant and the wild-type in terms of growth of hypocotyl, cotyledon, etc. in 2.5% agar that might differentially impact tension or CMT organization?

c) The use of propidium iodide to map cell separations is a key tool in this paper and should be better documented/validated. The authors should show higher magnification images of the bright staining (e.g., in Figure 3C, G and K) to convince the reader that this staining corresponds to cell-cell separations. This correspondence would be best demonstrated by showing a time-lapse of these stained regions separating (i.e., showing void between the disconnected cells) over time, thus directly demonstrating that these bright stripes indeed represent cell separations.

4) An explanation of results in light of previous literature is needed for to following issues:

a) Alterations to cell wall properties alters CMT organization and that interconnected feedback loops exist between with cell wall defects and CMT organization have been published. How might this alter the interpretation of the results?

b) The authors need to discuss how the tension-adhesion theory proposed here relates to the finding that only the innermost (most recent) cell wall layer determines the growth axis since adhesion is mediated by the middle lamella which is quite removed from the innermost cellulose layer.

c) The authors assert that "Here we provide evidence that the continuity of the outer cell wall is required for the coordinated response of adjacent cells to mechanical stress". How does this coordinated response differ from previously reported ones? Is the point that individual epidermal cells couldn't each respond to mechanical signals from the underlying tissue and create a seemingly coordinated response? I am not sure what a coordinated response would even look like if epidermal cells are missing/not next to each other-can you give an example of any coordinated response that doesn't involve continuity?

5) Can the authors say something about why the unadhered cells curl (e.g., Figure 1—figure supplement 1H) and not just grow straight? Presumably these cells are primarily feeling growth-based stress, so why do they curl?

6) The authors should take some caution about drawing strong conclusions about the competition between shape and growth based stress affecting microtubule rotary behaviors. Additional parameters such as wall heterogeneity could play a role in some cases. For instance, similarly shaped cells might differ in their wall thickness and/or composition depending on where they are in the hypocotyl which might change stresses, no?

Title: The title should be re-worded to make it more understandable to a wide audience. The word "nexus", which is used frequently in the text, could be replaced with "link" or "connection" to make this idea more accessible. For example, at the end of the Introduction, it could simply be stated that, "This mutant also allowed us to investigate how the loss of adhesion affects the propagation of mechanical stress and thus tension-dependent cell-cell coordination."

[Editors' note: further revisions were requested prior to acceptance, as described below.]

Thank you for resubmitting your work entitled "A tension-adhesion feedback loop in plant epidermis" for further consideration at *eLife*. Your revised article has been favorably evaluated by Christian Hardtke (Senior Editor), a Reviewing Editor, and two reviewers.

The manuscript has been greatly improved but there are a few remaining issues that need to be addressed before acceptance, these can be dealt with as text revisions relatively quickly as outlined below:

1) Reviewer 1 notes that the new description of cortical MTs in Figure 3 and Figure 5 and would like clarification about the relationship between their orientation in light of models of tension. The reviewer’s exact comments are below:

“However, I am puzzled about the relationship between Figure 5 and Figure 3. Are the CMTs different in light-grown hypocotyls (2.5% agar) between wild type and *qua1-1* mutants as expected if tension is disrupted by gaps?

In Figure 5, the argument is that CMTs are much less disrupted in *qua1-1* mutants in 2.5% agar compared to 1% agar (and that tension is maintained?) and that CMT organization is rescued. The CMT organization does look similar to wild-type light-grown hypocotyls. Was a direct comparison made?

Also in Figure 5: CMT orientation looked much more consistent in individual cells (and typically oriented in the same direction as neighboring cells) in 1% agar of *qua1* mutants compared to 2.5% agar. Perhaps there was more variability across the whole tissue, but between cells CMT organization looked coordinated. Perhaps this is due to loss of tension from gaps between cells?”

2) Stating that the *PMEI5* overexpression leads to homogenization of cell wall mechanical and chemical properties (as in the main text and Figure 3—figure supplement 2 legend) is a strong claim that would require more work to support it. The AFM measurements were not conducted over all cell wall surfaces. Also, one cannot exclude the possibility that the *PMEI5 OE* plants may have altered cell wall compositions through cell wall integrity pathways. Instead state that the *OE* plants have reduced heterogeneity of pectin esterification (as they did in the fifth paragraph of the subsection “Growth-derived stress dominates in hypocotyls”).

3) In the legend to Figure 1—figure supplement 5, fix to "closeups of panels B and D" not C. "Panel E also reveals the presence of threads…." not panels E and F.

---

## [Author Response]

Essential revisions:

1) The explanation for "reasoning that PMEI overexpression should induce more homogeneous wall properties in the hypocotyl" need to be explained more clearly and consistently and the overall rationale for this experiment explained. Also, in the conclusion from this experiment, couldn't this negative result just mean the experiment didn't work? Was the change in wall properties measured, or was there some other evidence that p35S::PME15 was working (some other phenotype that did occur?) The authors should discuss potential caveats of these overexpression experiments such as compensatory changes in wall composition/mechanics. Is p35S::PMEI5 the same as PMEI-OE?

The *p35S::PMEI5* seedlings indeed exhibit an altered phenotype. The original figure (Figure 3—figure supplement 2) actually shows twisted hypocotyls. Furthermore, two independent labs have shown how overexpressing PMEI5 affects cell elongation in the hypocotyl (Wolf et al., 2012; Müller et al., 2013) so we are confident this is a positive result. We have clarified these points in this revision to better account for the literature, phenotype and rationale behind this experiment.

2) In numerous places, experiments were not done on true qual/control pairs under the same conditions, or in same tissues. Below is a (non-exhaustive) list of places where the discrepancies should be addressed.

This is a fair point indeed, and we have included the required controls, as discussed below.

a) In the last paragraph of the subsection “A mechanical conflict at the petiole-blade junction of the cotyledon”, the authors mention a study done in leaves that they say is consistent with their results in cotyledons. It seems like a relatively straightforward experiment (given their current resources) to include an image of a leaf from qua-1 2.5% agar grown plants and to track the "cracks" using their pipeline to actually address this issue.

We performed the required experiment and found that the cracks in young leaves either followed the longitudinal axis of the leaf, especially at the leaf base as in cotyledons. However, these longitudinal cracks sometimes extended to a larger portion of the leaf base, consistent with the idea that growth in leaves is more anisotropic than in cotyledons. We also found more randomly oriented cracks, and interestingly enough, radial cracks around trichomes, which fits quite well with the predicted pattern of tensile stress in socket cells (see Hervieux et al., 2017). These new data are now presented in new Figure 4—figure supplement 2.

b) It is possible, perhaps even likely, that cell wall composition/structure is different between the blade and petiole cells. These differences might also contribute to the observed stress gradient at the blade-petiole junction. The authors should perform AFM analyses in this area to determine whether the cell stiffness and tension differ between petiole cells and their neighboring blade cells.

We performed AFM measurements at the blade-petiole junction and blade region on 2.5% agar (n = 10 cotyledons). Interestingly, the blade-petiole junction exhibited higher turgor, wall elastic modulus and apparent stiffness. Predicted tension was slightly higher at the petiole-blade junction than in the blade. These new results are consistent with the idea that there is a mechanical conflict at the petiole-blade junction. Although these data do not provide directional information, these data are also consistent with our claim that growth-derived tensile stress in higher at the petiole-blade junction. Although we cannot exclude that differences in wall polarities could also contribute to the gaping pattern in that region, our results on the *p35S ::PMEI5* line rather suggest that they would not be the primary cause of the gaps. In fact, cell wall mutants usually do not exhibit gaps. These new data are now included (new Figure 4—figure supplement 1) and discussed in this revision. Note that CMTs exhibit slightly higher anisotropy at the petiole-blade junction than in the blade, consistent with higher tension in that domain (see new Figure 4).

c) In Figure 2 it is unclear whether wild-type or qua1 mutants used for AFM analysis – both should be reported.

All AFM experiments have been performed on the WT, since we use *qua1* to understand the mechanics of the WT. Yet, an AFM analysis of *qua1* could indeed be interesting, also to provide an exhaustive characterization of our system. We found that *qua1* mutants (on 2.5% agar) exhibit lower turgor, softer walls and lower tension. It seems therefore that the mechanics of *qua1* reaches a scaled down, but similar, mechanical equilibrium as in the WT. Such a response may be consistent with the idea that *qua1* is less able to achieve epidermal continuity, and thus to build up pressure and tension in that layer. These new data are now presented in new Figure 2—figure supplement 1.

d) In Figure 2 different osmotic conditions used between experiment (2.5% agar) and AFM (water) measurements may cause alterations in turgor pressure because cells respond in minutes (Shabala and Lew, 2002). Does water immersion interfere with measurements?

In the long term, water immersion would affect the turgor status of the tissue. However, this effect should be small in the short term: single cell in cultures, epidermal peels and root epidermal cells (as in Shabala and Lew, 2002) may indeed respond in minutes to changes in osmotic conditions, but we find this response to occur much more slowly in the shoot. Typically, when Kierskowski et al. (2012, Science) use mannitol (0 to 0.4M) or NaCl (0 to 0.2M) on shoot apical meristem, they wait for 1 to 2 hours before recording an effect. Yet, to check for potential bias, we performed the same AFM measurements on cotyledons first submerged in liquid Arabidopsis medium (same as growth medium, only without agar), then changed to pure water, waited for 5min, and measured again (n = 6 cotyledons, n > 30 cells per condition). Almost all the samples were measured on the same region (same cells) before and after solution change. We found that the solution change did not affect turgor pressure P. In short, it seems that the measurement is sufficiently fast (and the differences in osmolarity, sufficiently small) to avoid major effects of the immersion solution on the measurements. We have included this new control in this revision (new Figure 2—figure supplement 1).

e) In Figure 2 use of different osmotic stabilizers between experiment (PEG) and AFM (mannitol) is also problematic. It would be more consistent to perform qua1 experiments in mannitol if the PEG is too viscous for AFM.

Mannitol can have dramatic effects on plant phenotypes (because mannitol is actively sensed by plants, Trontin et al., 2014 Plant J). This means that we cannot grow plants for a long time on mannitol without inducing many other effects. High molecular weight PEG is actually often preferred over mannitol to change osmotic conditions in the research field of abiotic stresses (e.g. Thoen et al., 2016 New Phytol.). We found that PEG was more suited for our phenotypic analysis.

f) In Figure 2 different cell types (cotyledon pavement cell) in AFM versus experimental material (hypocotyl, early cotelydon, petiole/leaf junction).

We now include AFM data for the petiole-blade junction in this revision (new Figure 4—figure supplement 1). As there is a shallow gradient of growth along the hypocotyl, we found that results were more reproducible in cotyledons to assess the impact of the growth conditions on tension. Also, the differences in cell wall properties likely are more homogeneous between cotyledon cells than between hypocotyl cells, given the bigger difference in cell size in the hypocotyl than in cotyledons.

g) Stiffness was measured on epidermal cells, not on the connections between cells, which are altered in the qua1 mutant. Does this reduction in stiffness reflect slower growth caused by growing in hyperosmotic conditions?

In *qua1*, the anticlinal walls are not accessible to the AFM because the top wall at that site is stretched as cells are separating, and masks the anticlinal wall beneath. In this revision, we include topographic and stiffness maps of WT and *qua1* intercellular junctions, showing the presence of softer wall fragments in *qua1* at the junctions. We also detect holes in these stretched/torn wall fragments, very similar to the ones observed by SEM in Bouton et al., 2002. These new data are now presented in new Figure 1—figure supplement 5. Note that we also measured the stiffness of *qua1* outer walls and found them to be softer, but as turgor pressure is also lower, the equilibrium appears similar to that of the WT, only scaled down (see new Figure 2—figure supplement 1). More generally, reduction in wall stiffness may not always reflect slower growth: softer walls would exhibit less resistance to turgor pressure, and thus allow faster wall deformation/yielding, and thus faster growth, assuming turgor remains constant.

h) Analysis of CMT organization (at the very top of the apex) in a region that does not match the longitudinal cell wall separation (lower down on the shank of the apex) (compare Figure 3C and 3D). Please perform CMT organization in the relevant part of the apex.

This has been corrected, we now show (and measured) CMT organization in the full stem apex in new Figure 3 and new Figure 3—figure supplement 3. The conclusions are unchanged.

i) Figure 3D compared microtubule organization in wild-type cells expressing GFP-MBD (in 1% agar?) to Figure 3C qua1 mutants in 2.5% agar. The agar concentration needs to be specified and also must be the same to compare the samples. Please perform your analysis of GFP-MBD in 2.5% agar or, if that was what was done, indicate that in the figure and legend.

We specified the agar concentration throughout and now show the results both in 1% and 2.5% agar concentration (see new Figure 3, new Figure 4 and new Figure 3—figure supplement 3). We observe the same trends in these two conditions, and thus our conclusions are unchanged.

3) More experimental detail should be presented for the following (some overlap with items in #2).a) In the fifth paragraph of the subsection “In the wild type, cortical microtubule orientations match tensile stress patterns, as inferred from cell adhesion defects in qua1” please briefly describe the ablation in more detail here in the text and also the methods (the only mention is a needle and a citation). How many cells are ablated? Is it a clean break? Is it only the epidermis? Can the tissue be imaged before this ablation, or only after?

We have reworded our protocol, with more details in this revision.

b) Is there a differential response between the mutant and the wild-type in terms of growth of hypocotyl, cotyledon, etc. in 2.5% agar that might differentially impact tension or CMT organization?

As mentioned above, we did not detect major differences in CMT behavior between 1% and 2.5% agar condition (see new Figure 3, new Figure 4 and new Figure 3—figure supplement 3). Interestingly, we could detect one small difference: we found that NPA-treated seedling exhibited less outgrowth at the stem apex on 2.5% agar than on 1% agar in the *p35S::GFP-MBD* line, which may suggest that decreasing the level of tension on 2.5% agar hinders organogenesis/outgrowth, which would nicely complement the results obtained by Sassi et al., 2014 or Fleming et al., (1997, Science), where outgrowth is promoted by modifying the mechanical properties of the wall.

c) The use of propidium iodide to map cell separations is a key tool in this paper and should be better documented/validated. The authors should show higher magnification images of the bright staining (e.g., in Figure 3C, G and K) to convince the reader that this staining corresponds to cell-cell separations. This correspondence would be best demonstrated by showing a time-lapse of these stained regions separating (i.e., showing void between the disconnected cells) over time, thus directly demonstrating that these bright stripes indeed represent cell separations.

To better describe the gaps, we acquired high-magnification images of PI-stained gaps in *qua1* at different stages of “gaping”. Interestingly, we could detect threads, likely from the outer wall being pulled apart, consistent with original SEM observations of *qua1* gaps in Bouton et al., 2002. We also include AFM images (topography and stiffness) of these junctions in this revision (new Figure 1—figure supplement 5).

4) An explanation of results in light of previous literature is needed for to following issues:a) Alterations to cell wall properties alters CMT organization and that interconnected feedback loops exist between with cell wall defects and CMT organization have been published. How might this alter the interpretation of the results?

CMT organization primarily affects cellulose deposition. In particular, the mechanics and adhesion of the middle lamella does not seem to be affected in any microtubule mutant. In other words, when CMT are aligned with tension, they might reinforce the cell, and the indirect effect of cellulose orientation on pectin and adhesion seems rather minor. Instead, cell-cell adhesion defects have been reported in actin mutants (e.g. *arp2,3*, Mathur et al., 2003), consistent with the role of actin in exocytosis and thus pectin delivery. Note also that we do not detect major CMT defects in qua1 mutant when grown on 2.5% agar, meaning that the CMT defects in qua1 on 1% agar is related to loss of adhesion, not the *QUA1* gene. We added this point in the Discussion.

b) The authors need to discuss how the tension-adhesion theory proposed here relates to the finding that only the innermost (most recent) cell wall layer determines the growth axis since adhesion is mediated by the middle lamella which is quite removed from the innermost cellulose layer.

Here we use CMTs as a proxy for tension patterns in tissues, and of course, tension controls more than only the CMTs. In our tension-adhesion theory, we would rather argue that the machinery that promotes adhesion (e.g. actin) would respond to tension to promote adhesion, as now included in the discussion. CMTs also align with tension, but seem to have little to do with adhesion. Somehow this recalls the observation that auxin transport through PIN1 seem to depend in part on tension, but independently from the CMT response to tension (Heisler et al., 2010 Plos Biol.).

c) The authors assert that "Here we provide evidence that the continuity of the outer cell wall is required for the coordinated response of adjacent cells to mechanical stress". How does this coordinated response differ from previously reported ones? Is the point that individual epidermal cells couldn't each respond to mechanical signals from the underlying tissue and create a seemingly coordinated response? I am not sure what a coordinated response would even look like if epidermal cells are missing/not next to each other-can you give an example of any coordinated response that doesn't involve continuity?

This comes down to stress magnitude. For instance, we (and others) propose that CMT align with maximal tension, whatever the cause is. If cells are separated, the maximum of tensile stress in that cell will be prescribed by pressure stress, i.e. cell shape (typically transverse in an elongated cell). If that same cell is under a thick outer wall where tensions can propagate and if the tissue is folding or if there are high differential growth between adjacent cells, then tension may be much higher than cell shape-derived stress and thus the local maximum of tension may be in a different direction than the one prescribed by cell shape. This is what was proposed for pavement cell shapes (Sampathkumar et al., 2014) and for cell division plane orientation for instance (see Louveaux et al., 2016). Our results in *qua1* formally show that this hypothesis is plausible, as now discussed in the paper.

5) Can the authors say something about why the unadhered cells curl (e.g., Figure 1—figure supplement 1H) and not just grow straight? Presumably these cells are primarily feeling growth-based stress, so why do they curl?

This comes down to the relaxation of stress when cells are separating. For instance, when dandelion stems are cut lengthwise, they curl up, because the epidermis is under tension and the inner tissues under compression (the contraction of the epidermis, which induces the curling, actually demonstrate that it was under tension in the uncut stem, see Kutschera and Niklas, 2007). Whether tension is caused by growth or shape, after cutting or *qua1*-dependent cell separation, the release of tension would cause a deformation, and curling is consistent with tissue tension. Hofmeister and later Sachs actually introduced the concept of tissue tension, and in their model, such tension was caused by the differential growth between epidermal and inner tissues. Our data, and other models, e.g. by Tobias Baskin or Oliver Jensen, are all consistent with that picture.

6) The authors should take some caution about drawing strong conclusions about the competition between shape and growth based stress affecting microtubule rotary behaviors. Additional parameters such as wall heterogeneity could play a role in some cases. For instance, similarly shaped cells might differ in their wall thickness and/or composition depending on where they are in the hypocotyl which might change stresses, no?

This is correct, especially as the CMT rotary behavior remains quite mysterious. It is also possible that local differences in turgor pressure may play a role in CMT rotations. We have toned down our statement.

Title: The title should be re-worded to make it more understandable to a wide audience. The word "nexus", which is used frequently in the text, could be replaced with "link" or "connection" to make this idea more accessible. For example, at the end of the Introduction, it could simply be stated that, "This mutant also allowed us to investigate how the loss of adhesion affects the propagation of mechanical stress and thus tension-dependent cell-cell coordination."

We have reworded the title and replaced nexus by “feedback loop” throughout.

[Editors' note: further revisions were requested prior to acceptance, as described below.]

The manuscript has been greatly improved but there are a few remaining issues that need to be addressed before acceptance, these can be dealt with as text revisions relatively quickly as outlined below:1) Reviewer 1 notes that the new description of cortical MTs in Figure 3 and Figure 5 and would like clarification about the relationship between their orientation in light of models of tension. The reviewer’s exact comments are below:“However, I am puzzled about the relationship between Figure 5 and Figure 3. Are the CMTs different in light-grown hypocotyls (2.5% agar) between wild type and qua1-1 mutants as expected if tension is disrupted by gaps?In Figure 5, the argument is that CMTs are much less disrupted in qua1-1 mutants in 2.5% agar compared to 1% agar (and that tension is maintained?) and that CMT organization is rescued. The CMT organization does look similar to wild-type light-grown hypocotyls. Was a direct comparison made?Also in Figure 5: CMT orientation looked much more consistent in individual cells (and typically oriented in the same direction as neighboring cells) in 1% agar of qua1 mutants compared to 2.5% agar. Perhaps there was more variability across the whole tissue, but between cells CMT organization looked coordinated. Perhaps this is due to loss of tension from gaps between cells?”

CMT organisations in wild-type and *qua1-1* seedlings grown on 2.5% look very similar indeed and we could not detect any significant difference in our analysis. As pointed out by the reviewer, we assume that this is due to the restoration of mechanical continuity of the epidermis. This is now included in the text:

“When grown on 2.5% agar medium, the wild type and *qua1-1* mutant hypocotyls exhibited similar CMT behaviors: we found a mean resultant vector length for *GFP-MBD* (2.5% agar) samples of 0.69 ± 0.15 and 0.65 ± 0.17 for *qua1-1 GFP-MBD* (2.5% agar) (*n*=606 cells in 11 samples and 865 cells in 15 samples, respectively) and the populations were not significantly different (*t*-test *p*-value = 0.556). This is consistent with the phenotypic rescue of the cracks in these conditions, which likely restores the mechanical continuity of the epidermis (Figure 5B, C).”

We have also clarified that our analysis of CMT coordination accounts for the whole tissue: CMTs look similar in WT and *qua1-1* at the single cell level; only their supracellular coordination is affected by the gaps in *qua1-1* (local disruption of the stress pattern). We have clarified this in the revised text:

“Whereas CMTs looked normal within individual *qua1-1* cells, they displayed less consistent orientation at the tissue scale in *qua1-1* cells than in the wild type (Figure 3L and 5A). […] In turn, the actual cell-to-cell coordination does not seem to be affected when two neighboring cells are still attached.”

2) Stating that the PMEI5 overexpression leads to homogenization of cell wall mechanical and chemical properties (as in the main text and Figure 3—figure supplement 2 legend) is a strong claim that would require more work to support it. The AFM measurements were not conducted over all cell wall surfaces. Also, one cannot exclude the possibility that the PMEI5 OE plants may have altered cell wall compositions through cell wall integrity pathways. Instead state that the OE plants have reduced heterogeneity of pectin esterification (as they did in the fifth paragraph of the subsection “Growth-derived stress dominates in hypocotyls”).

We have toned down the predicted impact of *PMEI OE* on wall properties as suggested by the reviewer.

3) In the legend to Figure 1—figure supplement 5, fix to "closeups of panels B and D" not C. "Panel E also reveals the presence of threads…." not panels E and F.

Thank you for spotting these errors – we have corrected the panel numbers.